# Cardioprotective Role of Captopril: From Basic to Applied Investigations

**DOI:** 10.3390/ijms26157215

**Published:** 2025-07-25

**Authors:** Marko Stoiljkovic, Vladimir Jakovljevic, Jovan Milosavljevic, Sergey Bolevich, Nevena Jeremic, Petar Canovic, Vladimir Petrovich Fisenko, Dmitriy Alexandrovich Tikhonov, Irina Nikolaevna Krylova, Stefani Bolevich, Natalia Vasilievna Chichkova, Vladimir Zivkovic

**Affiliations:** 1General Hospital “Sveti Luka”, 11300 Smederevo, Serbia; marko.stoiljkovic.md@gmail.com; 2Department of Physiology, Faculty of Medical Sciences, University of Kragujevac, 34000 Kragujevac, Serbia; jovan.milosavljevic1997@gmail.com (J.M.);; 3Center of Excellence for Redox Balance Research in Cardiovascular and Metabolic Disorders, 34000 Kragujevac, Serbia; 4Department of Human Pathology, I.M. Sechenov First Moscow State Medical University, 119991 Moscow, Russia; 5Department of Pharmacy, Faculty of Medical Sciences, University of Kragujevac, 34000 Kragujevac, Serbia; 6Federal State Autonomous Educational Institution of Higher Education, I.M. Sechenov First Moscow State Medical University of the Ministry of Health of the Russian Federation, 119991 Moscow, Russia; 7Department of Biochemistry, Faculty of Medical Sciences, University of Kragujevac, 34000 Kragujevac, Serbia; petar.c89@gmail.com; 8Department of Pharmacology, I.M. Sechenov First Moscow State Medical University, 119991 Moscow, Russia; vpfisenko@mail.ru (V.P.F.); tikhonov_d_a@staff.sechenov.ru (D.A.T.);; 9Department of Pathophysiology, I.M. Sechenov First Moscow State Medical University, 119991 Moscow, Russia; 10Department of Faculty Therapy, I.M. Sechenov First Moscow State Medical University, 119991 Moscow, Russia

**Keywords:** captopril, cardiac remodeling, oxidative stress, basic studies, clinical studies

## Abstract

Captopril, a well-established angiotensin-converting enzyme (ACE) inhibitor, has garnered attention for its cardioprotective effects in preventing heart remodeling and maintaining cardiac function, significantly improving life quality. However, recent studies have revealed that in addition to known hemodynamic alterations, captopril exhibits significant antioxidant, anti-inflammatory, and immunomodulatory effects that may underlie its protective mechanisms. Although it appeared to be overlooked in clinical practice, in recent years, additional efforts have been made to uncover the mechanisms of all drug effects, as recent research studies predict a wide spectrum of diseases beyond the recommended indications. This review thoroughly examines the mechanisms by which captopril mediates its protective effects, bridging basic biochemical observations with applied clinical investigation, especially during ischemic reperfusion (I/R) injury, hypertension, and heart failure (HF). Evidence points to captopril as a promising agent for modulating oxidative and inflammatory pathways that are crucial for cardiovascular medicine. Directions for future research are defined to determine the molecular targets of captopril further and to optimize its clinical utility in the management of cardiovascular and possibly other diseases.

## 1. Introduction

Captopril is an ACE inhibitor with a sulfhydryl group in its structure. It was the first animal toxin-based drug approved for human use since 1981. The development of captopril dates back to 1965, beginning with the separation of peptide fraction (Bradykinin-potentiating factor) from *Bothrops jararaca* venom, which potentiates the pharmacological actions of bradykinin both in vitro and in vivo [1]. Further research on animal models confirmed that the angiotensin-converting enzyme is inhibited by this peptide fraction [2,3]. As Cushman and Ondetti noted in 1991, captopril, the final product of these logical studies, is one of the simplest chemical structures as well as one of the most optimized drugs to be studied in the clinic for any indication [4]. Today, captopril is used to treat many cardiovascular diseases due to its bioavailability, pharmacodynamics, and kinetics [5,6,7]. Over the years, many scientists have sought to investigate the effects of captopril. These observations were extended to human trials and patients, as proven in experimental animals. Given that the cardioprotective effects of ACE inhibitors represent a well-investigated area, previous research experience in this domain by the co-authors of this review served as an additional incentive for the preparation of this article.

Undoubtedly, captopril, along with other ACE inhibitors, has found its place as a first-line therapy of arterial hypertension in leading guidelines [8] and a milestone in the treatment of HF with reduced ejection fraction (EF) [9], bringing about a major improvement in the therapy of these common health states. Notably, captopril’s application has extended beyond its primary indications, proving beneficial in various diseases and experimental conditions such as rheumatoid arthritis [10], experimental systemic lupus erythematosus-like disease [11,12], animal model of arthrofibrosis and osteoarthritis [13], proteinuria [14], alcohol-induced pancreatic fibrosis in rats [15], acute pancreatitis-associated lung injury [16], amiodarone-induced lung fibrosis [17], pulmonary hypertension in connective tissue diseases [18], experimental tumors [19], complications of insulin-dependent diabetes mellitus or model of induced diabetes mellitus [20,21,22,23], experimental autoimmune myocarditis [24], chemoprevention of hepatocellular carcinoma [25], experimental hepatic fibrosis [26], irradiation erythroid recovery [27], and Alzheimer’s disease [28], etc.

From the above, it can be inferred that in addition to cardioprotective effects, captopril also exhibits immunomodulatory properties by suppressing the production of pro-inflammatory cytokines such as tumor necrosis factor (TNF), Interleukin IL-1, IL-12 [29,30,31], antifibrotic properties due to the blockade of Transforming Growth Factor TGFbeta-1 overexpression or direct downregulation of TGFbeta-1 transcript [32], and other mechanisms of antioxidant and anti-proliferative activity, which will be explain below. As described in the mini-review, captopril also managed to halt the growth of various cancer cell types and to inhibit angiogenesis, exhibiting antineoplastic effects [33]. Recent data have revealed that it can even extend lifespan and control aging [34,35].

To address the complexity of this topic, the aim of this review article was to elucidate the molecular mechanisms and clinical implications through a coherent narrative and to reveal the breadth of captopril’s action. In addition to its well-known role in preserving cardiac structure and function, captopril also demonstrates various neurohumoral, antioxidative, immunomodulatory, antiarrhythmic, and antiatherosclerotic properties that contribute to the overall cardiovascular resilience. In summary, this review article furnishes a comprehensive analysis of findings regarding the cardioprotective effects of captopril, drawing on both experimental animal studies and clinical research in human populations. The discussion is structured around key hemodynamic, morphometric, and humoral changes to illustrate its effects. The pleiotropic effects of captopril in cardiovascular protection are illustrated in Figure 1.

## 2. Effects of Captopril on Cardiac Remodeling and Heart Function

Cardiac remodeling includes different morphometric and functional changes at macroscopic and cellular levels of myocytes and non-myocyte cells as a consequence of physiological adaptation and pathological conditions in response to cardiac load or injury. It embraces myocyte hypertrophy, cellular necrosis and apoptosis, and changes in fibrous tissues such as fibrosis, increased collagen content, and proliferation of fibroblasts [36].

A recent literature review indicated that the primary factors influencing ventricular remodeling following myocardial infarction (MI) include mechanical factors, such as increased afterload, preload, and wall stress, as well as biochemical changes, including different neurohumoral system activation, extracellular matrix remodeling, and altered cardiac metabolism [37].

### 2.1. Animal Models

Studies on animals have shown that continuous treatment with captopril, initiated after infarction, results in maintained or lowered left ventricular end-diastolic (LVED) pressure, reduced left ventricle volume, decreased left and right ventricular weight/body weight ratio (VW/BW), decreased end-diastolic volumes at rest and during the volume load, and increased EF compared with placebo controls or untreated animals in a rat models of acute MI [38,39,40,41,42,43,44,45,46]. In conclusion, administration of captopril contributed to arterial vasodilatation with a reduction in afterload. However, in the pathological state of HF, it also achieves venous vasodilatation. By obtaining the process of post-infarction ventricular remodeling, the study confirmed the effect of captopril in decreasing systemic systolic arterial pressure, preventing the increase in left ventricle (LV) and right ventricle (RV) filling pressures and RV systolic pressures. Heart weight/body weight ratio (HW/BW) was reduced significantly only in the group with a large MI compared to saline-treated rats, although the reduction was noticed in all groups irrespective of infarct size. Captopril also managed to prevent a decrease in minimal scar thickness as well as ventricular dilation and cardiac fibrosis [47]. In a canine model of MI, captopril has led to a decrease in mean left atrial pressure, as a measure of a preload, and in mean arterial pressure, with improvements in both regional and global systolic function [48]. It also reduced infarct size by decreasing both the area of myocardium at risk of developing necrosis and the amount of necrosis that developed in that area, via increased regional myocardial blood flow [49].

Unlike enalapril, captopril caused a greater decrease in infarct expansion and ventricular enlargement, with a superior improvement in EF, but a smaller decrease in collagen content in dogs with MI [50]. As noted, considering the varying results regarding the infarct-limiting properties of captopril [51], researchers investigated its effects in an open-chest pig model, given the greater similarity of the porcine vascular system to that of humans and the fact that myocardial injury has primarily been analyzed during the ischemic phase, but not during reperfusion. The results confirmed a reduction in myocardial injury upon reperfusion with captopril administration in a dose-dependent manner [52]. By comparing different ACE inhibitors, captopril significantly reversed increases in lactate dehydrogenase (LDH) release as a marker of cardiac damage induced by ischemia with a marked degree and duration of ACE inhibition ex vivo attributable to differences in distribution to key organs [53]. Appraising the HW as a parameter of hypertrophy [54] in morphological content, during the late captopril course after MI, the results do not align with the aforesaid outcomes since a significant HW reduction was not achieved; however, LVED pressure decreased, along with the mean circulatory filling pressure and increased venous compliance [55]. Similarly, captopril did not cause a regression of cardiac hypertrophy in the experimental model, even when administered immediately after ligation [56]. The major effects of captopril on hemodynamic and ventricular remodeling are presented in Table 1.

When captopril is initiated early after MI, it reduces hypertrophic growth of right heart chambers, but not left ones. Moreover, RV systolic and end-diastolic pressures were not changed significantly, while left filling pressures lessened with a minor improvement in LV systolic function and response to calcium [57]. Another study showed that prevention of right ventricular hypertrophy (RVH) during captopril started immediately after coronary artery ligation and continued for six weeks. Yet in the group treated during the first three weeks, only partial prevention of left ventricular hypertrophy (LVH) was observed, while late captopril treatment in both chambers led to reversed hypertrophic growth. Hence, the conclusion was that the RV is more sensitive to captopril than the LV [58]. These dissimilarities can be partially explained by a pattern of ACE density distribution, which is higher in atria than ventricles, and higher levels in right heart chambers compared to the left [59], along with increased concentrations of angiotensinogen mRNA in right heart chambers than left ones after dexamethasone [60] and greater ACE activity in RV walls than the interventricular septum [61].

However, besides preserving contractility and preventing decline in systolic function, extended captopril treatment not only limits LV remodeling slightly but also significantly mitigates the restrictive diastolic filling abnormalities observed in untreated rats with large MI, suggesting that changes in filling patterns due to captopril may determine changes in mitral flow patterns. As explained, alterations in preload, the different action of captopril on RV and LV, and changes in the myocardial passive properties would improve diastolic filling [62]. It can be concluded that captopril may confer some therapeutic benefits in the management of postischaemic HF through improvements in LV diastolic filling. Although the negative or unaltered chronotropic activity of ACE inhibitors is evidenced [39,56,57], unperceived results emerged from a study in which captopril led to an increase in heart rate (HR), which was explained as an autonomic compensatory reaction due to low values of aortic systolic and diastolic pressure [63].

Nevertheless, another study demonstrated that only high doses of captopril exhibit positive chronotropic activity, possibly through an adrenergic mechanism [64]. Neurohumoral activation due to MI is important for heart function and the process of remodeling [65]. As determined, neurohumoral activation, represented by levels of circulating neurohormones, increased steadily, comparable to infarction size. Besides increasing plasma renin activity, captopril has not significantly influenced norepinephrine, epinephrine, and aldosterone levels in rats with a sham, small, or moderate MI after four weeks of treatment. However, in a group with a large MI compared to saline-treated animals, captopril led to a significant decrease in plasma norepinephrine and epinephrine levels and a tendency to decrease aldosterone levels [47]. An increase in plasma renin activity has also been seen in treatment with perindopril, another ACE inhibitor, but without any influence on aldosterone levels [66].

**Table 1 ijms-26-07215-t001:** Major effects of captopril treatment on hemodynamic and ventricular remodeling.

Experimental Model	Dosage	Duration of Treatment (Follow-Up)	Effects	Suggested Mechanism	References
Rat model of myocardial infarction by coronary artery ligation	30 mg/kg/d i.p.	30 d	Lessening of the end-diastolic pressure and the muscle mass biventricularly	Reduction of the locally generated Ang II	[38]
Rat model of myocardial infarction by coronary artery ligation	2 g/L	3 m	Preservation of maximal forward output, lessens the ventricular dilation	Preload and afterload-reducing properties	[40]
Rat model of myocardial infarction by coronary artery ligation	2 g/L	3 w	Reduction in ventricle weight; shortening of the prolongation of the time to peak tension in papillary muscle; no changes in developed tension or passive stiffness; slight enhancement in muscle function	Improvement in loading conditions	[42]
Rat model of myocardial infarction by coronary artery ligation	2 g/L	5–6 w	Unchanged LV weight; decreased RV weight; reduction in LVED pressure and LVED volume	Decrease in blood volume and increase in venous compliance, combined with afterload reduction	[45]
Rat model of myocardial infarction by coronary artery ligation	2 g/L	1 w	Enhanced the contractile performance of spared myocytes; insignificant reduction in heart weight	Prevention of myocyte length and width increase	[46]
Dog model of coronary artery occlusion	0.25 mg/kg i.v. bolus +0.25 mg/kg/h	6 h	Reduced infarct size and area at risk	Improvement in collateral flow; reduction in afterload	[49]
Rat model of myocardial infarction by coronary artery ligation	2 g/L	5 w	Preserved contractility; limited systolic dysfunction; markedly lowered LV filling pressures; normalized the restrictive LV diastolic filling pattern	Decreased preload; reduction in RV-LV interaction; cardiac interstitium alteration	[62]

Although various studies have revealed the influence of MI on global ventricular function, its impact on contractility and the histology of spared myocardium remains obscure. Since it is noted that the papillary muscle’s stiffness correlates with that derived from stress–strain analysis of an intact normal ventricle [67] and that hypertrophy of the papillary muscles is similar to changes that occur in the free ventricular wall [68,69], researchers rely on it to investigate segmental active and passive properties of viable myocardium after MI. As presented, the mean myocyte cross-section area of the preserved papillary muscle of the LV is significantly increased in rats with large infarction, along with the myocyte diameter and a decrease in length at maximal developed tension, leading to myocyte hypertrophy, increased stiffness, and contractile dysfunction with increased collagen content. However, after a smaller infarction, minor functional changes are observed. During an evaluation of its isometric function and passive myocardial stiffness, the results showed that treatment with captopril attenuated the prolongation in time to peak tension and restored it to normal values; nonetheless, the developed tension, peak rate of tension increase (+dT/dt), persistent increase in myocardial collagen content, and muscle stiffness remained abnormal, implying that treatment with captopril alters the process of cardiac remodeling and hypertrophy and improves the parameters of contractility in noninfarcted myocardium [41]. Similar results are noted in the rat infarct model of HF, with only modest improvement in muscle function [42]. An insignificant decrease in myocyte cross-sectional area in morphological changes is also observed in the treated group compared to the untreated ones [41,42]. These data suggest that over the follow-up time of six weeks, postinfarcted HF tends to deteriorate over time, but with preserved muscle function at first.

Regarding histological changes, it is known that collagen fibers, as the most important component of the cardiac matrix, play an essential role in passive and active properties of the myocardium and, consequently, in the mechanical performance of the heart, despite the small fractional volume [70,71,72]. An increase in collagen content has been considered a major factor contributing to the development of HF [73]. It has been proven that collagen content increases in surviving myocardium after infarction [66,74,75,76]. This increase is thought to be related to the involvement of angiotensin II (Ang II) in post-infarction hypertrophy and reactive fibrosis. Supporting this, ACE inhibitors have been shown to decrease collagen content [66,76,77,78], while Ang II stimulates collagen synthesis in cultured rat cardiac fibroblasts [79] and incites chick cardiomyocyte protein synthesis and growth in vitro [80]. By assessing differential effects of Ang II on cardiac cell proliferation and intramyocardial perivascular fibrosis in vivo, infusion of Ang II into rats caused an increase in collagen deposition surrounding intracoronary blood vessels and an extension of collagen into the myocardial interstitium, with an increase in cell proliferation of adventitial and interstitial fibroblasts [81].

After conducting a study in which captopril initiated before induction of renovascular hypertension (RVHT) prevented hypertrophy, hypertension, and interstitial and perivascular fibrosis [82], a group of authors examined the influence of Ang II and aldosteron in RVHT. They assumed their possible pathogenetic origin in the development of myocardial fibrosis in RVHT, noting that both circulating Ang II and aldosteron lead to a significant rise in the interstitial collagen volume fraction and perivascular collagen area [83]. Ang II is recognized as a growth factor [84,85,86]. Following further studies, it was assumed that mechanical factors, a variety of chemical agents produced locally or systemically, such as bradykinin, prostaglandins, and transforming growth factor-beta, as well as the beta-adrenergic system, may also contribute to collagen content and fibroblast proliferation [87,88,89,90]. Collagen turnover and cardiac remodeling after MI are time-dependent processes [91,92], implying different times of captopril and other inhibitors of ACE administration.

As some results suggest, captopril therapy can prevent collagen proliferation and cardiac hypertrophy used either immediately after infarction or in an advanced phase of the remodeling process when it is almost developed. Another conclusion based on these data is that the RV is more sensitive to the effect of captopril than the LV due to higher angiotensin-converting enzyme activity in the right heart chambers [58,59]. Conversely, captopril therapy given at different periods following infarction showed that, whereas delayed therapy from 3 to 5 weeks after MI reduced afterload and improved cardiac performance in a dose-dependent manner, early therapy appeared to be even deleterious [93] due to inhibition of the cardiac remodeling processes by its inhibitory effect on increased DNA synthesis and the increased collagen formation [76]. As proposed, the negative effects of captopril administered during the early phases of remodeling are a consequence of the impact on normal adaptive responses of the heart due to the infarction [78]. This is confirmed by data in which captopril started immediately after MI inhibited endothelial cell proliferation and normalization of maximal coronary flow (MCF), yet late treatment had no influence on MCF [94].

Contrary to the previously mentioned studies, some researchers have found that myocardial collagen content remains abnormal despite treatment with captopril [41,42], while others show no effect on decreasing collagen level during late treatment [78]. One possible explanation for these findings is that a different methodology of collagen content evaluation was employed, as some authors declared that the epicardial and right chambers contain more collagen deposits [95,96,97,98]. Considerable data indicate the significance of collagen cross-linking present in myocardial remodeling as a potential target in the treatment of HF [99]. A paucity of significant differences between an early onset of therapy, 2 days after infarction, and delayed, 21 days after infarction [39,40], when the healing process is considered to be complete [100], insinuates slight effects of captopril when given in the early period of healing.

However, attempts to improve cardiac function should be made during the early stages of ventricular remodeling, as confirmed by further studies and trials [101]. The suppression of cardiac hypertrophy and enhancement in heart performance were associated with an attenuation of changes in myocardial gene expression due to early captopril administration. Results have highlighted the importance of ACE inhibition in normalizing the MI-induced gene expression of monoamine oxidase, cytochrome P450 2E1, cytosolic epoxide hydrolase, thrombospondin-4, latent TGF binding protein-2–like protein, and others, with explained improvements. However, captopril had no impact on the expression of enzymes included in cardiac energy substrate utilization or the expression of ion and water channels [102]. By considering the effects of early administration of captopril in spontaneously hypertensive rats (SHRs), authors showed a reduction in arterial blood pressure and LVH, explained by less ventricular expression of the atrial natriuretic peptide gene and *c-myc* gene [103]. Its permanent antihypertensive effect arose from alterations in the responsiveness of the brain renin-angiotensin system (RAS) in SHR [104]. These findings, along with others, indicate that use of captopril in the early phase of SHR life diminished the scope of cardiac hypertrophy and elevation in blood pressure with maintaining these effects well beyond drug discontinuation, or even permanently [104,105,106,107]. As opposed to the mentioned results, interruption of captopril soon leads to a progressive increase in arterial blood pressure, LV weight expressed through the LV/BW ratio, and parallel aortic remodeling [108].

The advantages of the timely use of captopril are indisputable and have been proved in many studies. However, a lesser number of results indicate the usefulness of captopril if it is applied in the later stages of the remodeling process. In a rat coronary ligation model, even a delayed treatment with captopril leads to an attenuation of myocyte remodeling and deceleration of chamber augmentation following infarction [109]. Beneficial effects of captopril on cardiac hemodynamic parameters were also seen in mice in a regional ischemia model when therapy was initiated one week after ligation [110]. Recent investigations utilizing 3D light sheet microscopy and echocardiography to assess cardiac function and morphological changes in a mice ligation model of MI demonstrated that delayed captopril treatment does not influence infarct size; however, it effectively prevents LV dilation and hypertrophy and improves border zone vascularization, resulting in enhanced EF [111].

In conclusion, there is no significant difference in the magnitude of changes between early and late effects of immediately started captopril therapy [39]. In summary, both early and delayed captopril administration resulted in discrepant data, as presented in Table 2.

Since matrix metalloproteinases (MMPs) have been recognized to play an important role in cardiac diseases and cardiac remodeling through degradation of the extracellular matrix, they have been the subject of many studies, which have been consolidated in various reviews [112,113]. Over the years, researchers have sought to evaluate the possible effects of the renin-angiotensin-aldosterone system (RAAS) on MMPs. In the hypertensive HF model, they ascertained the cardioprotective influence of the ACE inhibitor as it suppressed gene expression and gelatinolytic activity of MMPs independently of its depressor effect and prevented LV remodeling and dysfunction [114]. In the volume overload model of rat HF caused by aortocaval fistula and evaluating in vivo and in vitro MMP activity, it is shown that captopril significantly decreases MMP-2 activity in vitro, and prevents its increase in vivo in a concentration-dependent manner, independent of the renin-angiotensin system [115].

It has also been shown that captopril decelerates pulmonary arterial hypertension (PAH), and that remodeling develops in an aortocaval shunt [116]. To determine the relation between ACE inhibitors and MMPs in right heart chambers, RVH was induced with monocrotaline and the findings revealed suppressing effects of captopril on RVH, fibrosis, dysfunction, and reduced activities of MMP-2 and MMP-9 [117]. In the PAH model induced with pneumonectomy and monocrotaline treated with ACE inhibitor and AT1R blocker, administration of captopril and losartan alone results in a decrease in MMP-2 and MMP-9 expressions, which supports the previously mentioned effect of captopril. Another observation of this study was that these drugs delay pulmonary vascular remodeling [118]. Earlier, a group of authors suggested a positive effect of captopril and losartan on pulmonary vascular remodeling and attenuation of PAH [119]. However, in another model of pressure overload, RVH by pulmonary artery banding (PAB), captopril failed to improve morphological changes in the RV [120]. The administration of the AT1R blocker or captopril did not effect the extent of RVH induced by PAB, nor did it alter LV loading conditions [121]. To explain the discrepancy of results, authors offer different experimental models, which include different banding sites and severity levels, the type of RAS blockade, and uncertainty in the effect of RAS on load.

After the influence of RAS on cardiac remodeling was scrutinized in monocrotaline-induced RVH, it was necessary to test the hypothesis of RAS blockade attenuation of MMP-2 and MMP-9 expressions in β-adrenergic stimulation-induced cardiac remodeling with isoprenaline. Contrary to most research, this study demonstrated not only insufficiency of RAS blockade in attenuating MMPs expression and myocardial fibrosis, but significant augmentation in isoprenaline-induced MMP-9 expression due to captopril administration [122]. In order to evaluate the effects of the mineralocorticoid receptor antagonist and ACE inhibitor, since the influence of aldosterone on cardiac fibroblast and collagen content has already been established [79,123], spironolactone and captopril were used in the same model of cardiac remodeling, with captopril proving insufficient in preventing myocardial fibrosis, as seen in the aforementioned study, even in high doses, although both drugs decrease the VW/BW ratio [43]. In a rat model of primary and secondary hyperaldosteronism, spironolactone prevented fibrosis; captopril was unable to prevent it in primary hyperaldosteronism, but did prevent hypertension and LVH [124].

By assessing the endowment of RAS to isoproterenol-induced cardiac hypertrophy, another study noted that neither the AT1 receptor antagonist nor the ACE inhibitor was able to prevent the β-adrenergic stimulation-induced increase in ventricular weights [125]. These findings support the opinion that circulatory or cardiac RAS do not participate in cardiac responses to adrenergic stimulation of β receptors, although there is an indication that Ang II plays a role in maintaining but not inducing cardiac hypertrophy caused by isoproterenol [126]. Nevertheless, since the ornithine decarboxylase (ODC), as a rate-limiting enzyme in polyamine biosynthesis, included in cardiac hypertrophy induced by a β-adrenergic stimulation, is suppressed with both captopril and AT1 receptor antagonist [127,128], these results indicate that RAS may be involved in the induction of myocardial hypertrophy. Confirmation emerged from the results of a study in which captopril succeeded in preventing isoproterenol, caused an increase of polyamine contents, and led to a slight decrease in the HW/BW ratio during the late stage, which suggests that at least one phase of the polyamine cycle is RAS-dependent rather than the effect of direct β receptors stimulation [129].

In another model of adrenergic stimulation-induced cardiac hypertrophy but with norepinephrine, it was shown that chronic administration of captopril alone did not influence heart weights; however, when it is administered concomitantly with norepinephrine, a high, but not low, dose of captopril diminishes the development of LVH [130]. Different mechanisms of remodeling induction may explain discrepant results across models, as monocrotaline causes pressure overload remodeling of the RV, while isoprenaline stimulates β-adrenergic system with the LV as a subject of research. It is acknowledged that activation of β-adrenergic receptors stimulates renin release [131,132] as well as that aldosterone levels are significantly raised in isoproterenol-induced remodeling [133]. Following further results, it is claimed that aldosterone contributed to the profibrotic effect of Ang II in the heart [134], and that the cardiac fibrosis and hypertrophy stimulated by Ang II are related to activation of aldosterone synthesis in the heart [135]. Another finding was that captopril treatment led to upregulation of the β-adrenergic receptor in isoprenaline-treated neonatal rat cardiomyocytes by ACE inhibitor-mediated accumulation of bradykinin [136]. Furthermore, as elucidated, bradykinin induces proMMP-9 expression in rat brain astrocyte-1 [137]. Hence, it is surmised that captopril can promote MMP-9 expression due to the loss of bradykinin degradation by ACE or the increase in the cardiac β-adrenergic receptor. Another study confirmed an enhancement in the density and responsiveness of cardiac β-adrenergic receptors after captopril treatment [138].

In comparison with the isoprenaline-cardiac remodeling model, whereas captopril was insufficient to attenuate it, in the rat model of genetic hypertrophic cardiomyopathy, the drug has justified its cardioprotective role, both alone and in combination with low-dose spironolactone, by decreasing LVH [139]. The same drugs were evaluated in a hypertensive HF model in rats, whereas it is speculated that they could prevent end-stage congestive HF, probably through a renoprotective effect [140]. Earlier investigators found that AT1 receptor antagonists prevent RV fibrosis but enhance the increase in LV collagen content in isoproterenol-induced cardiac hypertrophy. It is therefore suggested that RAS has a different influence on different heart chambers [125]. In the pig model of progression of congestive HF caused by pacing-induced supraventricular tachycardia, the results implied a potential role of increased left ventricle MMPs along with time-dependent morphological and functional changes during the early phase of remodeling [141]. The effects of ACE inhibition on MMPs activity and, consequently, improved LV function and myocyte geometry were also described [142,143].

In light of thoracic aorta constriction (TAC) as a model of pressure overload-induced cardiac fibrosis and HF, captopril with pirfenidone significantly reverses deterioration of morphometric parameters and improves EF [144]. Further investigation confirmed the cardioprotective role of captopril, identifying its effect against TAC-induced cardiac apoptosis through the inhibition of the Wnt3a/β-catenin signaling pathway, and reducing myocardial hypertrophy by suppressing the Jak2/Stat3 pathway [145].

Contrarily, in subdiaphragmatic aortic-banded rats, treatment with captopril did not prevent the increase in the HW/BW or LV/BW ratios, and there was also no difference between the RV/BW ratios among treated and control groups. It was assumed that the LVH is blood pressure-dependent, whereas the changes in coronary artery medial wall thickening and perivascular fibrosis are dominantly mediated by the RAS, independent of changes in blood pressure. Supporting this, captopril therapy with large doses did not affect arterial blood pressure but lessened the increase in minimal coronary resistance, coronary artery medial wall thickening, and the increase in perivascular fibrosis seen in this rat model of pressure overload cardiac hypertrophy [146]. However, another study with the same model showed the efficiency of captopril in preventing myocardial cell hypertrophy and regression in myocardial cell hypertrophy as well [147]. Furthermore, findings indicated that captopril and losartan both attenuated pressure overload-induced depression of sarcoplasmic reticulum (SR) Ca^2+^ uptake and SR Ca^2+^-release activities, as well as both the Sarcoplasmic/endoplasmic reticulum calcium ATPase 2 (SERCA2) mRNA level and SERCA2 protein content [148]. By assessing the effect of ACE inhibitors as antihypertensive and renoprotective agents, it has been suggested that captopril has a direct effect on MMP-2 and MMP-9 activity in concentrations tantamount to those primarily required for inhibition of ACE in vivo. This effect appears to be mediated by its sulfhydryl group, as opposed to lisinopril, which lacks this moiety and demands higher concentrations to achieve the same level of inhibition [149]. From all the above, it can be concluded that the outcome of captopril therapy depends on the impact on positive and negative factors involved in the processes of cardiac healing and remodeling after a myocardial infarction.

Given the long-known impact of atherosclerosis and elevated blood lipid levels on the pathophysiology of cardiovascular diseases and mortality [150], captopril has been the subject of numerous studies. It has recently been revealed that it possesses antiatherosclerotic properties, manifested through the inhibition of dendritic cell maturation and the maintenance of their tolerogenic effects by promoting Treg polarization [151]. Earlier, it was proven that captopril therapy attenuates aortic lipid lesions [152,153] and exhibits an additive effect with statins in reducing the fatty streak area [154]. Additionally, captopril lowered total serum cholesterol in an obesity-induced hypertension model in dogs [155]. Observing the effects of captopril and valsartan, results show that both drugs exhibit a protective role in reducing and preventing the formation of atherosclerotic plaque in the arterial wall [156]. Captopril decreases atherosclerosis independently of LDL cholesterol and blood pressure by inhibiting macrophage-foam cell accumulation [157].

It is also crucial to mention the antiarrhythmic effect of early intervention with captopril during ischaemic-reperfusion-induced ventricular arrhythmia in various animal models, both in vitro and in vivo [158,159,160]. These protective effects of captopril on the inducibility of sustained ventricular tachycardia (VT) remain evident even two weeks after myocardial infarction [161,162]. The beneficial effects of captopril on reducing the incidence and duration of ventricular fibrillation (VF) after local ischemia and reperfusion were proven in another study [163].

Further confirmation was provided through an additional investigation comparing the effects of several ACE inhibitors, including captopril, enalapril, HOE 498, and its prodrug, on reperfusion arrhythmias in isolated rat hearts following 15 min of coronary ligation. In this study, researchers observed that captopril and HOE 498 prevented the occurrence of VF, with markedly reduced noradrenaline overflow during the initial minutes of reperfusion—properties which could not be attributed to enalapril and HOE 498 prodrug [164].

Another comparison described that captopril reduced the incidence of both reversible and irreversible VF, as well as mortality associated with irreversible VF, whereas losartan proved inadequate in achieving similar outcomes. Although both mentioned drugs decreased the occurrence of occlusion-induced VT, the reduction was not statistically significant. However, captopril did not reduce the incidence of reperfusion-induced VT, while losartan at doses of 2 mg/kg did. During the occlusion, neither drug resulted in a reduction in the incidence of ventricular premature beats. However, during the reperfusion period, both captopril and a higher dose of losartan effectively reduced the frequency of these arrhythmias. As discussed, kinins, prostaglandins, and NO, as well as major alterations in Ca^2+^ levels, also contribute to the antiarrhythmic effect of ACE inhibitors [165]. Despite the hemodynamic changes noted in this study, another comparative study indicated that, although the observed hemodynamic changes were not statistically significant, both drugs exhibited a comparable cardioprotective effect. This effect was evidenced by an increased VF threshold and reduced average episodes of VT and VF during both the occlusion and reperfusion periods, suggesting the hemodynamically independent antiarrhythmic effect of these drugs. Furthermore, in contrast to the previous finding, both drugs lowered mortality; however, this reduction was statistically significant only with losartan, and this discrepancy was explained by the authors via different routes of administration and experimental protocols [166].

To assess the effects of ACE inhibitors on noradrenalin release during ischemia and reperfusion, an evaluation demonstrated that captopril’s protective effect on vulnerability to VF was not mediated via alteration in catecholamine levels since the pattern remained unchanged throughout the whole protocol, conflicting with previous observations [167]. However, another suggested possible mechanism of ACE inhibition in these conditions can be explained through modification of local norepinephrine release via facilitation of prostacyclin synthesis and alteration in electrical stability of the potentiating action of Ang II on local elevated norepinephrine concentrations [168].

In summary, the mechanisms for these captopril actions were attributed to a reduction in cellular damage via reduced loss of high-energy phosphate nucleotides (purine), reduced noradrenaline overflow, improvement in contractility, increase in coronary blood flow, and inhibition of Ang II, an antiischemic effect associated with bradycardia and vasodepression. The significance of captopril is also evidenced by its ability to induce regression of ventricular hypertrophy, resulting in electrophysiological stabilization under both in vivo and in vitro conditions, thereby reducing the vulnerability to ventricular fibrillation [169].

As opposed to the described protective effects of captopril, other findings indicated that captopril possesses no direct antiarrhythmic activity at the average therapeutic dosage [170]. Earlier, electrophysiological research suggested that captopril’s antiarrhythmic actions are not mediated by established direct mechanisms in concentrations well above therapeutic levels [171]. The effects of captopril in the global ischaemic-reperfusion model of isolated rat hearts were also inconsistent with the aforementioned outcomes, since captopril could not influence the incidence of reperfusion VF or the frequency of spontaneous defibrillation. Additionally, no notable impact on noradrenaline overflow was observed. Although captopril successfully shortened the duration of VF with biochemical evidence of myocardial salvage, these results were insufficient to result in an improvement in measured hemodynamic parameters [172]. This disparity was explained through the protective effects of collateral blood flow observed in regional ischemia, a mechanism that is not present in the global ischemic model.

Numerous studies have established the role of RAS in cardiac preconditioning against I/R injury, offering different therapeutic modalities [173,174]. Researchers have assessed the contribution of captopril to preconditioning against MI, determining its capability in potentiating preconditioning through locally produced kinin and bradykinin B2 receptor activation. This cardioprotective effect occurs without elevating arterial plasma kinin levels [175]. Captopril, administered 24 h prior to the Langendorff protocols, also provides delayed protection against myocardial stunning in the globally ischemic rabbit heart through the activation of the same receptors. It preserves cardiac metabolism and coronary flow. Interestingly, all these effects were nearly completely abolished by the specific B2 receptor antagonist, except for the pro-arrhythmic effect seen in this study [176].

Another study, in which kininogen-deficient mutant rats were used in the coronary ligation model of MI, suggested that endogenous kinin possesses the capacity to reduce the size of MI via B2 receptor signaling due to enhanced regional myocardial blood flow around the ischemic area [177]. Investigating the effects of pretreatment with ACE inhibitors in a model of global ischemic arrest, captopril enhanced myocardial recovery after cardioplegic arrest and demonstrated greater efficacy compared to enalapril. These added protective effects of captopril were independent of blood flow, indicating that its cardioprotective properties extend beyond ACE inhibition-induced coronary vasodilation. As assumed, these effects may be associated with a thiol-dependent mechanism that limits oxidative damage [178]. A complementary explanation was proposed in a study on hypothermic myocardium, suggesting that captopril’s effects may be associated with its ability to scavenge free radicals, inhibit the production of Ang II, enhance the PGI_2_/TXA_2_ balance, and reduce calcium and sodium overload within the myocardium [179].

Captopril given in cardioplegia and in reperfusion exerted further protective effects on the recovery of isolated rat hearts, improving hemodynamic performance, coronary flow, and oxygen consumption [180]. Similarly, acute administration of another ACE inhibitor during cardioplegic arrest improves postischemic hemodynamic data, such as systolic and diastolic function, coronary perfusion, and high-energy phosphates levels in a rat model of HF [181]. Captopril further justified its cardioprotective properties in another study, in which a four-week course of captopril enhanced myocardial resistance to ischemia, as evidenced by reduced LDH release into the coronary effluent during postischemic reperfusion. Furthermore, captopril lowered myocardial lactate levels at the end of reperfusion. It is proposed that the decreased lactate accumulation seen in captopril-treated groups was due to improved lactate washout, attributed to enhanced recovery of coronary flow. Another notable finding was the increase in glycogen levels and a trend toward elevated ATP content in perfused hearts under basal conditions, suggesting an enhanced energy reserve [182]. Conversely, captopril does not exert significant effects on myocardial function or adenylate energy metabolism, but enhances coronary flow during normoxic perfusion [183].

In addition to all the aforementioned cardioprotective effects of captopril, it is important to underscore that long-term captopril administration improved survival in a rat model of MI [184,185].

### 2.2. Human Investigations

As results originating from animal-based experimental models have postulated, the effects of short- and long-term captopril treatment need to be acknowledged in human trials. It has been concluded that administration of captopril in patients with HF decreases systemic and pulmonary vascular resistance, improves cardiac index, and causes a fall in mean arterial pressure and pulmonary-capillary wedge pressure, leading to a decrease in the cardiothoracic ratio, improvements in exercise tolerance and symptoms, and reduced hospital days and requirements for hospital admissions [186,187,188,189,190,191,192,193,194,195,196]. A controlled trial of captopril in patients suffering from chronic HF with NYHA class II-IV revealed that during acute treatment response, at rest, the hemodynamic findings are consistent with previously reported results. However, despite this, exercise capacity did not change significantly since exercise duration, maximal workload, and maximal oxygen consumption were congruent both before and after captopril. During upright exercise, captopril slowed the increase in LV filling pressure and led to a moderate increase in cardiac and stroke indexes, but less than expected given the results obtained at rest. However, after 3 months of captopril, hemodynamic changes were sustained, and exercise capacity was increased [197].

As a possible explanation for the lack of more intense exercise hemodynamic changes seen in this study, the authors suggested the timing of the exercise measurements, which were not always at the time of the peak drug response. A year earlier, by examining the effects on radionuclide measurements at rest during the acute response to captopril, it was noticed that both the left and right ventricular EF were increased, mainly due to reductions in the end-diastolic volume and ventricular filling pressure, since an insignificant change in stroke volume was observed. Hemodynamic data throughout exercise were similar to the results at rest, but it is worth noting that the decrease in LV filling pressure after captopril was less intense, while the increase in stroke volume was statistically significant as opposed to values at rest. Nonetheless, the level of exercise after acute administration of captopril remains unchanged [198].

Delayed effects and dissociation between hemodynamic and exercise response have also been observed in other vasodilators, such as hydralazine and isosorbide dinitrate [199,200]. Researchers elucidated this by the intensity of peripheral oxygen utilization and the inability of the muscle vascular bed to dilate properly during exercise, since the abnormalities in the regional circulations accompanying congestive HF have been proven earlier [201]. By evaluating regional blood flow redistribution in HF following captopril, despite a significant decrease in total systemic vascular resistance and pulmonary vascular resistance as well as an increase in cardiac index, hepatic and forearm flow do not increase, nor does vascular resistance in both regions decrease [202].

Contrary to these results, estimating peripheral circulatory dynamics found that captopril administration increases forearm blood flow and causes diminished forearm vascular resistance and forearm venous tone, with the suggestion that captopril produces more venous than arteriolar dilating effects [194]. Regardless of the improvement in cardiac output, splanchnic dilatation did not occur, but renal blood flow increased by 60% after captopril administration in HF [203]. Augmentation of cerebral blood flow has also been reported [204]. Patients with HF have a different model of blood flow redistribution as forearm, splanchnic, and coronary flow remain unaffected by captopril treatment, whereas renal blood flow increases [205]. In animal models of HF with preserved EF and metabolic syndrome and chronic kidney disease as comorbidities, captopril exhibits overall renoprotective effects since it increases renal blood flow and glomerular filtration rate along with improved renal histology and reduced proteinuria [206]. Despite improvements in systemic hemodynamics, results from another short-term captopril therapy differed since it did not improve renal hemodynamics due to the distinct hypoperfusion observed in this study, with a decrease in coronary sinus blood flow but without affecting coronary sinus oxygen saturation [207].

In the manner of long-term treatment, continuous improvements in symptoms and tolerance to both treadmill stress and bicycle tests, along with sustained hemodynamic effects, were noted after 8 weeks or more of captopril, with maintenance after an average follow-up length of almost 2 years [186,187,188,189,195,196,208]. In addition, with a mean follow-up of approximately 6 months, improvements in clinical status and hepatic and kidney function were also noticed [190]. A sequential hemodynamic study with an overall follow-up period of half a year revealed a gradual improvement in symptoms and hemodynamic changes in congestive HF over time due to long-term captopril therapy, with the most steady response in shortened pulmonary mean transit time and significantly reduced cardiopulmonary volume [209]. Other monitored parameters are in accordance with the results of the aforementioned studies, confirming the arteriolar and venous vasodilatory effects of ACE inhibitors. To determine the predictive value of short-term hemodynamic changes induced by captopril in patients with stable HF NYHA II-III, researchers were unable to find significant correlations between these measurements and the subsequent changes in clinical class, exercise tolerance, EF, and cardiothoracic ratio after 3 months of therapy [210]. However, in patients with severe congestive HF with functional class NYHA III-IV, it is ascertained that pretreatment and post-treatment stroke work index, as well as post-treatment stroke volume index and cardiac index, were predictive of the long-term clinical response to captopril. The stroke work index has also been recognized as a major determining hemodynamic factor of survival [208]. The hemodynamic and neurohumoral effects of short-term and long-term captopril administration are summarized in Table 3 and Table 4.

In the SAVE trial, long-term administration of captopril in MI survivors with asymptomatic LV dysfunction resulted in lessened cardiovascular mortality and morbidity and improvements in survival [212], with reduced recurrence of MI and need for revascularization [213]. The antiischemic effect of captopril is also observed, along with improvements in systolic function and prevention of LV dilatation after myocardial infarction [214]. The same investigators confirmed the prevention of LV dilatation and LV diastolic dysfunction, along with an improvement in EF [215]. In patients suffering from MI who were symptomless, with EF below 45%, treatment with captopril did not result in any increase in diastolic volume but led to a significant reduction in systolic volume while stroke volume index and EF were significantly increased, similar to the aforementioned studies [216,217].

The effects of captopril on patients with definite Q wave MI without clinical evidence of HF were verified, as the results showed a reduction in the LV end-systolic volume index and augmentation of EF, with a statistically nonsignificant rise in LVED volume index. The conclusion drawn was that captopril prevents ventricular dilation typically observed after MI when initiated 24–48 h after symptom onset, emphasizing the importance of prompt treatment [218]. Specifically, administration of captopril within 24 h of acute MI affirmed the enhancement of LV function and prevented the progression of ventricular dilatation seen in patients with both impaired and relatively preserved LV function, highlighting captopril’s efficacy across a range of initial cardiac performance following MI [219]. Notably, a major benefit for regional wall motion from very early captopril administration 2 h after thrombolysis occurred in patients with anterior infarction and either TIMI 0–2 flow or a corrected TIMI frame count greater than 27 [220]. Earlier, it was assumed that the simultaneous early application of intravenous captopril and recombinant tissue-type plasminogen activator during acute MI preserves LVED volumes and pressures compared to the placebo group [221]. After anterior MI in patients with EF ≤45%, use of captopril led to less LV filling pressure and increased exercise capacity [222].

The Munich Mild Heart Failure Trial revealed that the conjunctive effect of captopril, along with standard therapy for HF, slowed the progression of HF, especially in the NYHA I-II class [223]. The Captopril Multicenter Research Group provided a trial in which they evaluated treatment with captopril in patients with chronic HF refractory to digitalis and diuretics. The results highlighted benefits in terms of symptoms, increased exercise tolerance, improvement in NYHA classification, and EF during the 12 months of adjunctive captopril administration. It was also observed that the mentioned improvement was insignificantly greater in baseline NYHA III-IV and in those whose HF is caused by primary myocardial disease as opposed to ischemic etiology [224]. As evidenced during acute captopril intake in patients with chronic congestive HF poorly controlled by digitalis and diuretics, hemodynamic benefits result from the inhibition of Ang II production, accompanied by a reduction in sympathetic activation and an enhanced release of vasodilatory prostaglandins, including PGE_2_ [225].

In contrast to other vasodilators used in congestive HF, such as hydralazine and prazosin, captopril leads to a greater decrease in systemic resistance and mean arterial pressure as well as a rise in cardiac index. Captopril was also found to significantly reduce plasma aldosterone levels, while the reduction in plasma catecholamines was not statistically significant [211]. Given the known involvement of sulfhydryl groups in vascular smooth muscle in the mechanism of action of organic nitrates [226], it is assumed that captopril, thanks to its chemical structure, can potentiate the hemodynamic effect of nitrates. This was also proven in a study that compared the influence of captopril, ramipril, as an ACE inhibitor without a sulfhydryl group, and cysteine as a sulfhydryl-containing agent on the vasodilatory effect of isosorbide dinitrate [227].

To provide information for clinical practical guidelines, controlled trials have been conducted. The ACE Inhibitor Myocardial Infarction Collaborative Group contributed to this through a systematic review of short-term trials such as ISIS-4, CCS-1, CONSENSUS-II, GISSI-3, SMILE, GISSI-3 Pilot, ISIS-4 Pilot and several smaller studies in which ACE inhibitor treatment with captopril (in ISIS-4, CCS-1), enalapril (in CONSENSUS-II), lisinopril (in GISSI-3), and zofenopril (in SMILE) was stared early after MI and continued up to six weeks [228]. In conclusion, with several limitations due to the unavailability of individual patient data from the SMILE study, smaller studies that cannot be included along with the aforementioned larger studies because of the number of randomized patients, which was under 1000, and variations in administration routes, dosages, pharmacokinetics, and concomitant medications, it is observed that early treatment with ACE inhibitors leads to approximately five lives saved per 1000 patients treated for around one month. The most significant advantages occurred within the first seven days, during which the mortality risk is at its highest. The proportional reductions in mortality were not significantly different between patients with prior comorbidities, HF, or KILIP class >1 at entry, as opposed to those without it or KILIP class 1, but absolute benefits of ACE inhibitor treatment were greater among those groups, which implies that patients at higher risk generally benefit to a greater absolute extent.

The long-term randomized trials SAVE, AIRE, and TRACE, which involved the administration of different ACE inhibitors—captopril, ramipril, and trandolapril—in patients with LV dysfunction or clinically manifested HF, were later combined by the same collaborative group in a systematic review of MI [229]. As noted, the initiation of ACE inhibitors between 3 and 16 days after MI onset resulted in a rapid reduction in mortality within a few weeks of the start of treatment, which tended to increase with the duration of treatment. During another long-term randomized placebo-controlled study, which included patients with severe HF of different etiologies, classified as NYHA III or IV research, confirmed a significant improvement in hemodynamics and clinical parameters due to captopril therapy. This was due to marked changes in noninvasive parameters, such as a reduced radiological heart volume and significant hormonal changes, with a decrease in noradrenalin and an increase in plasma renin, while the decrease in adrenaline and aldosterone was not significant [230]. Within humoral changes observed during therapy with captopril, many of the previously mentioned studies also proved its effects on decreasing aldosterone levels [189,231,232]. Regarding plasma alterations, it is seen that captopril leads to an increase in norepinephrine levels, which is explained via direct or reflexive neurogenic stimulation but without physiological significance [189]. In opposition to this, results confirmed lower norepinephrine levels after captopril administration [203].

Contrary to these results, an initial decrease in aldosterone levels was observed after one month of captopril therapy in hypertensive patients, followed by a significant late increase. Aldosterone levels returned to their control values after 3 to 6 months and surpassed these values after one year of treatment; this was probably not dose-dependent since dose adjustment was not made after 9 months of treatment, but plasma aldosterone levels continued to rise [233]. This phenomenon, known as aldosterone escape, is described in the review, offering possible explanations based on the increase in Ang II and potassium seen during chronic administration of ACE inhibitors in different cardiovascular states, and additionally mentioning ACE inhibitor resistance due to increased aldosterone levels [234]. The magnitude of aldosterone escape is variable since a range from 56 to 1568 pmol/L of aldosterone level has been seen in patients with severe HF treated with captopril [235]. In summary, the results reveal controversy surrounding this escape as no evidence of hormonal escape was observed during long-term captopril therapy [236].

Considering observations that myocardial production of Ang II can persist despite treatment with captopril [237] due to incomplete blockade of Ang II production during ACE inhibition, and production of it independent of ACE via chymase [238], the OPTIMAAL trial was conducted to compare the effects of ACE inhibitors and AT1 receptor antagonist son mortality and morbidity in patients suffering from MI with evidence of HF or LV dysfunction. These results, investigated in the context of the ELITE II trial, strengthen the conclusion that captopril treatment should remain the preferred treatment over AT1 receptor antagonists due to the overall results favoring captopril for patients with HF in the absence of adverse events [239,240]. The VALIANT trial compared the effects of captopril and another AT1 receptor antagonist, valsartan, either combined or alone. Valsartan, as a clinically approved alternative, proved to be at least as effective as captopril in reducing the risk of major cardiovascular outcomes after myocardial infarction. However, combining both drugs led to an increase in the rate of adverse events without improving survival, indicating that the choice between these alternatives should be individually clinical assessed [241].

However, another comparison of conventional antihypertensive regimens, which include β blockers and diuretics with captopril in the Captopril Prevention Project (CAPPP) study group, showed that the cardiovascular mortality was modestly lower in captopril-treated patients, but fatal and non-fatal stroke was more common with captopril. This may be attributed to their elevated baseline and study-period blood pressure measurements, alongside a more frequent history of stroke and transient ischemic attacks, compared to those receiving conventional treatment. Notably, captopril was more effective in the prevention of diabetes mellitus [242].

Following further research, analysis of the class effectiveness of various ACE inhibitors after MI revealed that different ACE inhibitors show similar clinical efficacy regarding long-term outcomes such as mortality and reinfarction when used in comparable dosages. Although captopril users initially displayed a higher risk of reinfarction, this difference was eliminated following dosage adjustment. Additionally, used dosages of ACE inhibitors administered in clinical practice are lower than those utilized in clinical trials, especially for captopril, as the most underdosed ACE inhibitor in this comparison. Interestingly, after a follow-up of 3 years, crossover between the used ACE inhibitors was observed, which was highest for patients using captopril. However, among patients not switching agents, no differences in clinical outcomes between the different exposure groups were found. Another conclusion was that the association between ARBs and clinical events was similar to ACE inhibitors, further supporting the clinical equivalence of these drug classes in the management of MI survivors [243].

Although findings from the CATS study are in line with those from the SAVE trial [212], in light of a reduced number of ischemia-related events, these effects became significant only after 3 months and were maintained 9 months following captopril therapy, though no improvement was observed during exercise testing. Notably, discontinuation of treatment led to a high incidence of ischemia-related events [244]. In the manner of exercise tolerance, captopril administered in a single dose has been shown to increase the maximal working time (MWT) and maximal workload sustained (MWS), to reduce S-T depression at maximal rate/pressure product (MRPP), and to significantly prolong the time needed to reach the maximal blood pressure, HR, and MRPP in normotensive patients experiencing effort-induced angina pectoris [245].

Additionally, sublingual administration of captopril improved the parameters of the maximal exercise test [246]. Similarly, captopril, when used as a standalone antianginal agent, enhanced exercise tolerance in patients with coronary artery disease [247,248]. Although captopril administration did not result in any significant improvement in exercise duration to the onset of angina or the progression to moderate angina severity when compared to placebo, combination with isosorbide dinitrate resulted in more expressed antianginal and antiischemic effects [249]. However, in patients with coexisting HF and angina pectoris, treadmill-exercise time on a high-intensity protocol was shorter with captopril than with placebo, and was associated with increased angina symptoms [250].

The antiarrhythmic effect of captopril seen in animal models has been extended to human research, which confirmed that early administration of captopril within 24 h in MI reduces the incidence of late potentials in the heart as a marker of electrical instability and risk for malignant ventricular arrhythmia (VA) [251]. In a randomized placebo-controlled trial of oral captopril and mononitrate treatment, early initiation of captopril in MI resulted in a reduction in ventricular extrasystoles and the frequency of multiple extrasystoles over 48 h of Holter ECG monitoring, though these effects did not reach statistical significance when compared to the placebo group [252].

The need for a direct investigation into whether these agents enhance both early and long-term survival through a reduction in VA prompted the randomized, placebo-controlled substudy trial ISIS-4, in which captopril treatment led to a significantly lower mean number of ventricular ectopic beats and a significantly lower number of patients with frequent VAs demonstrating antiarrhythmic effects in both early and late phases of MI. However, there were no significant differences in the incidence of ventricular tachycardia or of complex VAs [253]. Captopril, started 7 days after the onset of MI, did not reduce the occurrence of couplets per VT after a month and three months, but after six months it succeeded in expressing antiarrhythmic effects that correlate with the LVED volume index and myocardial ischemia as an independent predictor of VA. In summary, captopril has been shown to have a beneficial impact in patients with LV dysfunction after MI, reducing both the number of complex VAs as well as the number of patients developing VAs during the chronic phase [254]. In the thrombolytic era, pretreatment with captopril prior to urokinase administration resulted in a reduction in early ventricular hyperkinetic arrhythmias (VHA) as well as predischarge VAs, compared to initiating captopril three days after thrombolytic therapy [255].

Similar to these results, captopril administered before thrombolysis resulted in a lesser occurrence of early and late VHAs [256]. Additionally, findings from the Captopril And Thrombolysis Study (CATS) corroborated the antiarrhythmic properties of captopril, showing a lessen number of patients experiencing paired ventricular premature beats, accelerated idioventricular rhythm, and non-sustained VT at all-time points compared to the placebo group, with significantly fewer patients in the captopril group experiencing non-sustained ventricular tachycardia and accelerated idioventricular rhythm during the acute thrombolytic phase in MI [257]. Evaluating short- and long-term effects of beta blockers alone or in combination with captopril during thrombolysis in patients with anterior MI, captopril exhibited a significant reduction in the incidence of early ventricular arrhythmias while its effect on late arrhythmias was not significant [258].

In a small double-blind trial involving patients with severe congestive HF, 24-h ambulatory ECG monitoring showed that treatment with captopril reduced the incidence of ventricular arrhythmias, including ventricular extrasystoles, couplets, ventricular salvoes, and VT [235]. Comparing the effects of therapy between captopril and digoxin in patients with mild to moderate HF during established diuretic therapy, researchers confirmed the protective effects of captopril in reducing ventricular premature beats (VPBs) in patients with more than ten VPBs per hour at baseline compared to the digoxin group, but not the placebo [259].

As reviewed in animal models in which the role of Ang II in cardiac preconditioning against I/R injury was demonstrated [260], a human model of preconditioning revealed that administration of captopril and lisinopril improved postischemic functional recovery through B2 receptor activation and bradykinin production [261]. The beneficial effects of captopril on myocardial protection during cardioplegia in coronary artery bypass graft (CABG) surgery were explained through notably reduced CK and norepinephrine levels as well as a reduction in myocardial ACE activity, which did not reach statistical significance [262].

Given the previously mentioned impact of MMPs on cardiac remodeling in animal models, further research focused on investigations involving human tissues. A study performed on human HF caused by ischaemic and idiopathic dilated cardiomyopathy confirmed the inhibitory effect of captopril and ramiprilate on MMP-2 and MMP-9 in vitro [263]. The same drugs were used to investigate their inhibitory effect on MMP-2, considering that activation of MMPs can lead to autolysis of gelatinases under more acidic conditions. Since MMP activity is known to be sensitive to pH changes, this effect is thought to result from the observation that ACE inhibitors may cause a drop in solution pH at millimolar concentrations, potentially having a negative impact on enzyme activity. As concluded in this study, administration of captopril and lisinopril in concentrations found to inhibit MMP-2 are three orders of magnitude higher than those present in vivo conditions; the results therefore showed the absence of direct inhibitory effect of these ACE inhibitors on MMP-2 in concentrations reached in vivo [264]. However, research performed in patients on continuous ambulatory peritoneal dialysis (CAPD) therapy revealed that captopril directly inhibited MMP-2 activity in peritoneal effluents in a dose-dependent manner at micromolar doses from 1 to 100 µmol/L [265].

## 3. Role of Oxidative Stress and Circulating Molecules in Captopril-Induced Cardioprotection

Oxidative stress presents a disturbed balance between reactive oxygen species (ROS) production and the endogenous antioxidant defense mechanisms, resulting in excessive ROS concentration linked to multiple pathophysiological pathways in the heart, as summarized in reviews. As described, at low levels, ROS possess the ability to regulate differentiation, proliferation, and excitation–contraction coupling in the heart under physiological conditions, while at high levels they cause oxidation and damage to DNA, proteins, and other macromolecules leading to cardiomyocyte dysfunction, apoptosis, contractile dysfunction, impaired cardiac remodeling, fibrosis, hypertrophy, and, consequently, HF. Since the role of antioxidants has been recognized, different enzymatic and non-enzymatic endogenous as well as exogenous antioxidants have been described [266,267,268,269].

By examining the influence of D-penicillamine as a slow-acting antirheumatic drug, in 1985 scientists discovered that captopril, as another compound with a thiol-containing group, had a similar antioxidant superoxide dismutase (SOD)-like activity [270]. Considering that captopril has a sulfhydryl (SH) group in its structure, which was previously proven to be responsible for the scavenger activity of glutathione (GSH), researchers have established captopril as a potent free radical scavenger [271]. Comparing its activity with already known scavengers, results showed that captopril scavenged superoxide anion (O_2_^−^) as effectively as SOD, hydroxyl radical (OH·) as effectively as dimethylthiourea (DMTU), and hypohalite radical (OCL·) and hypochlorous acid (HOCL) better than allopurinol. The effects on the free radicals generated by activated polymorphonuclears were even more impressive since they were equivalent to the combined effects of SOD, catalase, and allopurinol [272].

Investigating the in vitro antioxidant power of various ACE inhibitors using the ferric reducing antioxidant power (FRAP) assay, captopril exhibited significantly higher antioxidant activity compared to other non-sulfhydryl-containing ACE inhibitors [273], refuting an earlier investigation that disputed captopril’s antioxidant properties [274]. Using electron spin resonance (ESR) with 5,5-dimethyl-1-pyrroline-N-oxide (DMPO) as a spin trap, the results indicated that both SH and non-SH-containing ACE inhibitors scavenge OH· more strongly than O_2_^−^. It was concluded that the free radical scavenging activity of ACE inhibitors is not exclusively related to the presence of the SH moiety but also to the deesterification reaction, since the potency of non-SH ACE inhibitors is significantly reduced by deesterification [275].

The attenuation of reperfusion-induced myocardial dysfunction in vitro by captopril is believed to be due to its scavenging activity, which functions independently of ACE inhibition, and is attributed to the presence of the SH group [276], known to possess an antioxidant activity [277]. Comparative studies evaluating ACE inhibitors with varying chemical structures demonstrated improvements in postischemic contractile function and reduced cell death. The findings indicated that these beneficial effects were exclusive to ACE inhibitors containing an SH group, such as captopril and zofenopril, confirming their role as scavengers of OH· and OCL·, and showing their ability to inhibit the enhanced lipid peroxidation observed during reperfusion. Although significant differences in the postischemic myocardial membrane phospholipid composition were not observed, captopril and zofenopril effectively reduced the contents of nonesterified fatty acids, including palmitic, linoleic, oleic, and arachidonic acids [278,279]. A general free radical scavenging activity, rather than a specific superoxide scavenging activity, was observed for the mentioned SH-containing ACE inhibitors [280]. It was later confirmed that SH agents preferentially scavenge non-superoxide anion radicals. Captopril was found to scavenge other toxic oxygen species, including hydrogen peroxide (H_2_O_2_) and singlet oxygen (^1^O_2_), though the biological significance of these actions remains unclear, while it scavenges concentrations of HOCL that are pertinent under physiological conditions. Additionally, it inhibited microsomal ascorbate-induced lipid peroxidation and reduced O_2_^−^ production in zymosan-stimulated granulocytes. Non-SH-containing ACE inhibitors also inhibited microsomal ascorbate-induced lipid peroxidation, but less so [281]. Upon reevaluation, it was found that captopril possesses no inherent ability to scavenge superoxide radicals directly [282].

Another study confirmed that the SH moiety, as well as enalapril, do not scavenge superoxide radicals in vitro, either in human neutrophils stimulated with f-MLP or in the xanthine–xanthine oxidase reaction [283]. Both research groups concurred that the purported superoxide radical scavenging effect of captopril was misinterpreted because SH-containing compounds, such as captopril, can reduce the spectrophotometric absorbance of cytochrome c, thereby leading to a decrease in the superoxide-induced reduction of cytochrome c. Another in vitro evaluation revealed that captopril is a weak scavenger of O_2_^−^, but a potent scavenger of OH· and OCL·, with the ability to decrease lipid peroxidation, as evidenced by reduced malonaldehyde (MDA) formation [284], confirming previous results in which captopril scavenged O_2_^−^, OH·, and HOCL plus OCL· along with significant inhibition of arachidonic acid levels and MDA seen during reperfusion [285]. A reduction in MDA has also been observed in erythrocytes of type II diabetic patients treated with captopril independently of the GSH level [286]. Despite prior findings supporting the antiperoxidative effects of captopril, results from an in vitro lipid oxidation model suggested its ineffectiveness as a peroxyl radical scavenger activity even when used at concentrations exceeding therapeutically achievable levels [287].

An additional assessment of captopril’s efficacy in scavenging ROS demonstrated that captopril is a potent scavenger of HOCL. If it reacts at all with superoxide and H_2_O_2_, the reaction occurs slowly. Captopril reacts rapidly with OH, but despite this, during therapeutic use, it is unlikely to express significant potency due to the low concentrations observed in vivo. Although captopril demonstrates its OH scavenging activity, it does not significantly inhibit the iron ion-dependent generation of OH from H_2_O_2_ or lipid peroxidation stimulated by iron ions and ascorbate, or by the myoglobin-H_2_O_2_ system. Notably, in the presence of ferric ions without ascorbate, captopril can even stimulate lipid peroxidation [288].

Other findings supported the pro-oxidant action of captopril peroxidation, when iron is the main catalytic metal, through the formation of OH and thiyl radicals (RS), which have the potency to initiate lipid peroxidation [289]. In the presence of ferritin, captopril acts as a pro-oxidant [290]. This is not surprising since thiols can initiate lipid peroxidation via a direct reduction of iron, by inhibiting endogenous antioxidant mechanisms, and generating RS [291,292,293]. Further supporting this notion, another lipid peroxidation study revealed that, at lower concentrations, captopril expresses its pro-oxidant properties similarly to the aforementioned findings. Another conclusion was that it is a less effective antioxidant compared to GSH, suggesting that captopril’s cardioprotective properties are likely mediated through non-radical mechanisms rather than direct free radical scavenging. Additionally, captopril’s beneficial effects in mitigating reperfusion injury may also not stem from direct free radical scavenging, as results have shown that although captopril reacts rapidly with hydroxyl radical and the thiocyanate radical anion, its efficacy in this capacity is relatively modest [294].

Animal hearts treated with captopril 10 days prior to I/R injury exhibited better recovery, especially when reperfused with captopril after arrest. As discussed in the last study, a major role in the prevention of I/R injury is assigned to Na^+^ K^+^ and Ca^2+^/Mg^2+^ ATPase enzymes, and the results revealed that captopril acted as a free radical scavenger which restored Na^+^ K^+^/Mg^2+^ ATPase and Ca^2+^ Mg^2+^/ATPase levels in I/R injury [295], confirming previously established findings that long-term treatment with captopril as enalapril induced a significant increase in Na^+^K^+^ and Ca^2+^–Mg^2+^ ATPase enzymes activities in erythrocytes membrane in hypertensive individuals [296]. Similar previous findings have also indicated that oral pretreatment with captopril and captopril-enriched reperfusion solution is significantly effective in improving post-ischaemic myocardial function and decreasing myocardial injury. Another observation was a correlation between increased lipid peroxidation and depletion of myocardial GSH content [297].

Defining the cardioprotective role of SH and non-SH-containing ACE inhibitors during MI in rats, researchers have found that ACE inhibitors can protect the heart against I/R injury and these effects of captopril were attributed to the decreased production of Ang II due to ACE inhibition and the increased release of myocardial prostacyclin (PGI_2_) [298]. Other findings indicated that captopril enhanced further prostaglandin 6-keto PGF_1α_ and thromboxane TxB_2_ production during reperfusion of the ischemic myocardium, as well as inhibiting arachidonate and MDA formation, suggesting that it promotes the cyclooxygenase pathway and the prostaglandin synthesis through its direct free radical scavenging activity [285]. Contrary to these results, captopril had no significant effect on an overflow of these cyclooxygenase products along with PGE_2_ in isolated rat hearts [299].

Another study demonstrated the potency of captopril to induce PGI_2_ synthesis in vascular tissue [300]. It was discovered that indomethacin, as a cyclooxygenase inhibitor, abolished the PGI_2_-mediated protective effects of ramiprilat, while it did not completely abolish these effects of captopril, implying additional anti-free radical activity related to the SH group of captopril [301]. As ascertained, PGI_2_ and its analogs also possess antioxidant activity [301,302,303,304]. Since changes in plasma concentrations of Ang II and bradykinin seem insufficient to fully explain the hypotensive effect of captopril, scientists assumed the presence of other mediators of the vascular response to captopril and discovered that on both high- and low-sodium diets, captopril caused a significant dose-related increase in the 13,14-dihydro- 15-keto metabolite of the vasodilatory PGE_2_, which correlated with a decrease in diastolic blood pressure. However, no significant changes were observed in the plasma levels of 6-keto PGF_1α_ or TxB_2_ as the stable products of PGI_2_ and TxA_2_, as opposed to the aforementioned findings [305].

Confirmation of the increase in PGE_2_ metabolite levels due to captopril administration was obtained through another study, which further concluded that pretreatment with indomethacin, which causes a blunted depressor response to captopril, provides evidence that prostaglandins are an important component of captopril’s mechanism of action, especially in the sustained phase of response, while early response to captopril could be related to the Ang ll and kinin changes [306]. Another study, involving pretreatment with indomethacin, indicates that changes in prostaglandin production expressed through an increase in PGE_2_ metabolite levels play a significant role in mediating the hypotensive response to captopril, both in the acute and chronic phases. This major role of prostaglandin is particularly pronounced in groups of patients with a high salt intake [307]. In patients with low-renin hypertension, the antihypertensive effects of captopril are attributed to the induction of the release of endogenous vasodilating prostaglandins, while in normal renin hypertension, a prime antihypertensive mechanism is based on inhibition of the renin-angiotensin system [308,309]. Contribution of PG to the antihypertensive effects of captopril is supported by other studies, which suggest interactions and clinical importance when anti-inflammatory drugs that inhibit PG-synthetase are used concomitantly with captopril [310]. The role of PG in the hormonal and hemodynamic effects of captopril has been confirmed in another study [311]. When investigating eicosanoid synthesis of platelets in SHR during in vitro captopril administration, researchers found that captopril significantly reduced thromboxane synthesis in platelets of SHR with lower blood pressure, while there was no difference in thromboxane generation in platelets of the SHR with higher blood pressure. However, captopril increases the level of PGD_2_ and 12-L-hydroxy-5,8,10-heptadecatrienoic acid, thereby inducing vasodilatation [312].

From the above, it can be elucidated that captopril-induced changes in eicosanoid synthesis in SHR platelets may represent an additional mechanism contributing to the drug’s antihypertensive effects. It can also be assumed that these effects on the platelet arachidonate cascade are likely not related to captopril’s ACE inhibition, but rather due to the presence of the free sulfhydryl group of the drug. Other mechanisms by which Ang II induces hypertension in SHR are through oxidative stress and endothelin, since captopril treatment lowered mean arterial pressure, Ang II, oxidative stress, as indicated by thiobarbituric acid reactive substances (TBARS), and endothelin in SHR. However, in WKY, on the contrary, it did not significantly impact these variables. Notably, both groups exhibited increased plasma renin activity [313].

When evaluating the effects of thiol-containing captopril and non-thiol-containing enalapril on hypertension in SHR, both drugs demonstrated hypotensive properties, with captopril exhibiting a more pronounced effect. Despite both treatments enhancing nitric oxide synthase activity in the heart and aorta to similar levels, neither agent was able to increase endothelial nitric oxide synthase (eNOS) expression. Additionally, conjugated diene concentrations, as a marker of membrane oxidative damage, were reduced following treatment with both drugs; however, in the aorta, this reduction was significantly greater in the captopril-treated group [314].

Conflicting results indicated that captopril did not cause significant activation of NOS in any of the tissue homogenates of the heart, aorta, brain, or kidney in vivo [315] as well as in vitro in vascular smooth muscle cells [316]. Similarly, captopril did not influence eNOS protein expression [317]. These findings suggested that cardioprotective effects of captopril achieved through ACE inhibition may occur via mechanisms independent of enhanced nitric NO production. Further investigation revealed that while captopril inhibited the onset of hypertension induced by L-NAME, primarily by mitigating the heightened sympathetic vasoconstriction response, antioxidant N-acetyl cysteine enhanced NO-dependent vasodilation without affecting sympathetic tone and with only modest antihypertensive potency [318]. However, NO plays a role in the acute hypotensive effects of captopril [319]. Furthermore, both captopril and enalapril increased nitric oxide production [320].

An in vitro study evaluating the effects of SH and non-SH-containing ACE inhibitors on the endothelium under chemically induced oxidative stress via the generation of superoxide and hydroxyl radicals suggested direct hydroxyl radical scavenging and antiperoxidative activity of SH agents [321]. Another observation substantiated free radical scavenging activity and lipid peroxidation-inhibiting action of captopril compared to non-SH-containing ACE inhibitors [322]. Furthermore, research supported the protective effect of captopril on the endothelium against ROS related to superoxide anion scavenging activity. As opposed to the aforementioned findings, however, this effect has also been attributed to enalapril [323]. Additionally, it has been demonstrated that captopril can prevent the homocysteine and advanced oxidation protein products-induced inhibition of endothelium-dependent relaxation in isolated rat aorta, possibly through scavenging oxygen free radicals and increasing NO production in endothelial cells [324,325,326,327]. Although the last study highlights captopril’s potential in counteracting oxidative stress in endothelial dysfunction induced by homocysteine thiolactone in a dose-dependent manner, losartan and enalaprilat can only partially prevent inhibition of endothelium-dependent relaxation due to homocysteine thiolactone.

Although enalapril exhibits stronger ACE inhibiting activity, captopril demonstrates greater protective effects in preventing the inhibition of endothelium-dependent relaxation in mice subjected to a high-methionine diet, while maintaining SOD activity, NO level, and attenuating the decrease in paraoxonase PON1 activity with an evidenced decrease in serum MDA concentration [328]. However, another study concluded that both captopril and enalapril not only alleviated this inhibition but also counteracted the increase in MDA concentration and the decrease in SOD and glutathione peroxidase (GSH-Px) activity in serum induced by exogenous advanced oxidation protein products in rats, implying that these protective effects of captopril are not related to the presence of SH moiety [329]. Contrary to these results, neither of the drugs managed to enhance SOD and GSH-Px activity in endothelial cells; rather, they even decreased the activity of GSH-Px. MDA concentration remained unchanged [330].

Despite the mentioned mechanisms by which captopril contributed to the improvement in impaired endothelium-dependent relaxation by homocysteine, a reduction in endogenous asymmetric dimethylarginine (ADMA) as an inhibitor of nitric oxide synthase was described. Another observation concerned a reversion of serum MDA, nitrite/nitrate levels, and hepatic SOD activity [326]. Although both captopril and cysteine, which contain SH groups, were shown to significantly reduce H_2_O_2_ levels when exposed to induced reactive oxidative environments, enalaprilat, lacking an SH group, had no effect on it. Interestingly, while captopril exhibited stronger scavenger activity, specifically within the oxidant-generating system, cysteine was more effective in preventing oxidant-induced cell injury at equimolar concentration. Neither cysteine nor captopril altered intracellular GSH levels. As concluded, despite the described protective effects in oxidant-related endothelial and renal tubular epithelial cell injury, cell injury was not prevented, even in doses that exceeded therapeutic levels, suggesting captopril’s weakness as an antioxidant in biological systems [331].

It is notable that captopril maintains its protective effect on endothelium-dependent responses in the SHR even after withdrawal [332]. Contrary to these findings, earlier it was observed that, despite the sustained antihypertensive effect of captopril following the discontinuation of treatment, endothelium-dependent relaxation remained impaired in captopril-withdrawn SHR [333]. The authors suggested that these differences may be attributed to the age at which captopril treatment was started.

Given that summarized reviews have shown that redox-active transition metal compounds can catalyze Fenton-type and Haber-Weiss reactions, leading to the production of ROS and causing damage to lipids, proteins, and DNA [334], researchers aimed to evaluate the effects of captopril in these metal-catalyzed ROS generation processes. The results confirmed that, besides its free radical scavenging activity, captopril can also chelate metals, expressing SOD-mimicking activity via the formation of the thiol/copper complex [289,335,336]. However, the antioxidant effects of captopril were disputed since it is shown that it did not affect free radical production when metal ions were not required as catalytic agents, but its protective role has rather been attributed to its ability to chelate metal ions included in the autoxidation process [337]. SOD activity was also seen in the captopril–iron complex [338].

Furthermore, beyond its free radical scavenging, captopril exerts its antioxidant effect as a recyclable free radical scavenger, suggesting it can keep GSH in a reduced form [339]. Contrary to the above, captopril caused a shift to a more oxidized glutathione pool. It is also excluded that it acts by increasing GSH synthesis [340].

Investigation into the molecular mechanisms involved in the regulation of oxidative status in SHRs revealed captopril’s ability to upregulate antioxidant enzymes and eNOS gene expression, as well as to downregulate p22phox expression in the heart since chronic therapy increased levels of GSH-Px, glutathione reductase (GSH-Rd), and SOD in erythrocytes of SHRs, occasionally surpassing values observed in normotensive WKY rats [341].

Distinguishing features of myocardial I/R injury include pronounced oxidative and nitrosative stress, coupled with excessive accumulation of intracellular calcium Ca^2+^ [342]. Another described mechanism by which captopril enhances its cardioprotective effects is through the preservation of sarcoplasmic reticulum (SR) Ca^2+^-pump activities and the tendency to preserve membrane protein SH group content [343,344]. An opposing explanation was that captopril primarily targeted the Ca^2+^ release channel rather than the Ca^2+^ pump, as evidenced by its lack of effect on the Ca^2+^-ATPase activity in cardiac SR vesicles uncoupled by alamethicin. Captopril facilitated Ca^2+^ reuptake by reversing sulfhydryl oxidation of ryanodine receptors (RyRs), thereby restoring the closed state of the Ca^2+^ release channels. Complementing this, results have suggested that captopril diffuses into myocytes and serves as an effective myocardial protectant against ischemic injury by inhibiting the oxidation of hyperreactive protein thiols [345].

The advantages of captopril treatment on cardiac Ca^2+^ uptake and Ca^2+^ pump ATPase activities have also been observed regardless of whether treatment is administered early or late following the coronary artery ligation. Notably, captopril prevented the decrease in SERCA2 and phospholamban protein contents and mRNA levels [346]. Alterations in MI-induced SR Ca^2+^ transport have also been described when some other ACE inhibitors like trandolapril and imidapril were administered [347,348]. Captopril’s contribution to the enhancement of overall antioxidant cellular capacity is further reflected through the increased activity of endogenous antioxidants such as GSH, GSH-Px [349], and SOD-1 in hypertensive patients [350]. However, SOD and GSH-Rd remained high compared with the healthy controls [349].

It is worth mentioning the different effects of captopril on organ-specific enzymatic activity. Authors have shown that captopril increased liver CuZn-SOD, Mn-SOD, and Se-GSH-Px activities during 4–11 weeks of treatment. The same was observed for enalapril, albeit at a slower pace [351]. Another investigation involving brain, kidney, and heart tissue homogenates of mice revealed that captopril increased CuZn-SOD activity in the heart and kidney medulla, whereas enzyme activity in brain, renal cortex, and erythrocytes remained unaffected, as did Mn-SOD and Se-GSH-Px across all examined tissues. Enalapril, on the other hand, enhanced the activity of CuZn-SOD in the renal medulla, heart, and erythrocytes and Se-GSH-Px activity in the renal medulla. Catalase activity was not affected by either of the treatments. Additionally, by measuring tert-butyl hydroperoxide-initiated chemiluminescence (HICL), captopril improved non-enzymatic antioxidant properties in the brain, while enalapril increased protection in both the brain and kidney medulla [352]. Further investigation reported that both captopril and enalapril, with doses corresponding to those prescribed for humans, enhanced total glutathione content (GSSG+GSH), Se-GSH-Px, and GSSG-Rd activities in various mouse tissues.

In enalapril and captopril-treated mice, an increase in GSSG+GSH was observed in the brain and erythrocytes, with the distinction that enalapril also exerted effects on the lungs and increased Se-GSH-Px activity in the renal medulla, whereas captopril did not affect these tissues. However, Se-GSH-Px activity was increased in the liver and kidney cortex by both drugs. Regarding GSSG-Rd, both drugs increased their activity in erythrocytes, brain, and liver, but differed in terms of their impact on tissue specificity and magnitude since enalapril augmented activity in the renal cortex, and captopril resulted in a greater increase in activity in the liver. The activity of GSSG-Rd was unchanged in heart, lung, and kidney medulla tissue homogenates. Interestingly, neither of the drugs affected either the plasma level of α-tocopherol or ubiquinol-9 as lipid-soluble antioxidants. Another conclusion was that captopril, like enalapril, lowered TBARS production in erythrocytes, as opposed to a previous study in which enalapril did not affect it, while captopril inhibited TBARS production in liver and skeletal muscle homogenates. As noted, these differences concerning organ specificity may derive from different chemical structures, tissue metabolization, and penetration of captopril and enalapril [320].

In contrast, intraperitoneally administered captopril produced a dose-dependent depletion of hepatic GSH, with comparable depletion seen in orally treated mice [353]. A time-dependent and dose-dependent decrease in hepatic GSH has also been seen in mice and rats [354]. Notably, the doses of captopril used in these studies far exceeded therapeutic levels, implying that it is unlikely that captopril would cause a significant reduction in hepatic GSH at therapeutic doses. Additionally, chronic administration of captopril in mice did not result in any cumulative effects on hepatic GSH depletion. Further investigation revealed that during I/R injury, it is crucial to administer captopril prior to ischemia due to a lower extent of lipid peroxidation expressed through prevented TBARS production, which is not seen in captopril after treatment. Myocardial GSH, catalase, and SOD activity showed no significant changes during therapy [355]. Despite exerted cardioprotection against I/R injury, similar to previous findings, captopril did not affect GSH-Px and catalase enzyme activities but showed an increase in the activity of SOD [356].

However, additional research concluded that, whereas different antihypertensive drugs, captopril, hydralazine, and terazosin reduced blood pressure in both SHR and WKY rats, notable alterations occurred in the specific antioxidant enzyme activities and lipid peroxidation levels. These changes varied according to both the tissue type and the rat strain. Furthermore, in SHR rats, antihypertensive drug treatment led to more significant modifications in antioxidant enzyme activities compared to those observed in WKY rats [357]. The results indicated that all three drugs led to an increase in myocardial Cu/Zn-SOD activity and a decrease in GSH-Px activity in SHR. Captopril did not significantly affect Mn-SOD activity in the myocardium; however, it did result in a reduction in Mn-SOD and GSH-Px activity in the liver. Captopril was the only agent that resulted in a decrease in catalase activity within the liver, while simultaneously exhibiting no effect on myocardial catalase activity. Regarding catalase activity in the myocardium, captopril treatment resulted in an increase in WKY rats, whereas hydralazine and terazosin did not alter catalase activity, in contrast to the reduction seen with these drugs in SHR. Although similar activities in GSH-Px were seen in the liver and myocardium in both strains of rats, captopril did not show any effect on GSH-Px activity in the WKY myocardium. Lastly, myocardial TBARS levels were reduced by all three drugs in WKY rats, whereas only hydralazine decreased these levels in SHR rats. As illustrated in Table 5, organ-specific enzymatic activity observed with captopril, in comparison with other antihypertensive agents and ACE inhibitors of non-SH moiety, merits further attention.

Evaluation of different antihypertensive and antilipemic treatments in rats with induced non-alcoholic fatty liver disease revealed a decrease in MDA concentration in the captopril group. Conflicting results showed a decrease in the activity of both GSH-Px and GSSG-Rd, but increased α-tocopherol concentration as well as PON activity in the liver during captopril administration [358]. Assessment of oxidative stress showed an altered enzyme pattern of GSH-Px in patients with essential arterial hypertension and therapy with captopril and enalapril did not significantly influence the activity of erythrocyte Se-GSH or Se-non-dependent GSH-Px, or SOD activity after one year of treatment [359]. Earlier, it was concluded that erythrocyte lipid peroxides, erythrocyte catalase activity, and oxidative stress in erythrocytes did not show significant differences between the two drug study groups at baseline, nor were any significant changes observed during or after the six-month treatment period [360].

Findings that underline the benefits of using combined antihypertensive therapy describe that the combination of atenolol and captopril offers superior protection against oxidative stress compared to the administration of either drug individually. This dual therapy helps maintain serum GSH and MDA levels closer to those observed in normotensive individuals, suggesting enhanced efficacy in reducing oxidative damage [361]. It must be emphasized that concomitant administration of other substances and drugs enhances the synergistic cardioprotective and antioxidative effects of captopril [362,363,364,365].

Although the pro-oxidative properties of captopril have been previously addressed, it is essential to conclude that captopril should not be exclusively regarded as a drug with significant antioxidant potential, given the findings of studies demonstrating its pro-oxidative effects, particularly when it interacts with redox-active transition metals like iron, and especially copper [366,367]. A concise overview of the captopril dual role in oxidative balance is outlined in Table 6.

Despite early recognition of the immune system’s role in cardiovascular disease pathogenesis, research in this field continues to advance. Key immunological mechanisms involved in both physiological and pathological processes have recently been described, including complex signaling pathways and the engagement of both innate and adaptive immunity, highlighting the potential for immune-targeted therapies [368]. Investigating the immunomodulatory activities of captopril, researchers have found conflicting results.

As suggested, captopril therapy may present benefits for patients with chronic HF since it has been shown to decrease TNF-α levels [369]. Another finding corroborates this observation, noting that captopril administration did not alter IL-1β or IL-2 levels, which were not assessed in the previous study [370]. It is shown that captopril decreases inflammation in viral and drug-induced myocarditis by reducing oxidative stress and DNA damage, as well as altering cytokine profile [371,372]. Additionally, captopril reduced cardiac inflammation in SHR through inactivation of NF-κB-dependent pro-inflammatory factors such as IL-1β and IL-6, while increasing the anti-inflammatory cytokine IL-10 [373]. In chronic HF induced by isoproterenol, captopril, either alone or combined with melatonin, reversed cytokine changes in TNF-α, IL-6, and IL-10 [374]. Treatment with atorvastatin, losartan, and captopril in individuals with coronary artery disease and hypertension led to upregulation of TGF-β and IL-22, and downregulation of IL-6 [375,376]. Moreover, in human coronary artery endothelial cells exposed to high dextrose, captopril reduces the secretion of pro-inflammatory cytokines [377].

The anti-inflammatory potential of captopril was demonstrated in a study in which, at concentrations exceeding those used in clinical practice, it succeeded in reducing IL-1β levels by as much as 76% [378]. Furthermore, captopril enhances the levels of IL-2 and IL-10 [379]. It is also demonstrated that it can decrease immune-complex deposition in the brain and IFNα levels [12]. As previously discussed, in addition to inhibiting dendritic cell maturation and promoting Treg polarization, captopril also increased IL-10 and TGF-β and decreased IL-6 and IL-12, exhibiting antiatherosclerotic activity [151]. Captopril also inhibited lipopolysaccharide (LPS)-induced production of cytokines TNF-α, IL-1α, IL-10, IL-12, and IL-18 from dendritic cells [380]. However, despite a dose-dependent decrease in pro-inflammatory cytokine concentrations in vitro, anticytokine activity of captopril in vivo was disputed [381]. This stands in contrast to earlier results demonstrating such effects in both in vitro and in vivo settings, although large doses were required [382]. In light of the increasing body of evidence supporting the role of myeloperoxidase (MPO) in inflammation, atherosclerosis, and acute coronary syndrome [383], its interaction with captopril has been investigated. As evaluated in isoproterenol-induced cardiac damage, pretreatment with captopril decreased the cardiac MPO activity [384]. Similarly, a captopril-induced reduction in MPO was noted in another study, proving its cardioprotective effect [385]. When addressing the impact of captopril on various cell populations, studies have demonstrated significant immunomodulatory effects. It has previously been shown that captopril may induce a suppressor activity involving monocytes and the T_8_ lymphocyte subset [386]. Investigating the effects of captopril and diuretics on the Th1 cell- and macrophage-mediated cellular immune response, researchers found that it lowered the hapten-presenting activity of macrophages and inhibited secretion of IL 12 [387]. It enhances the humoral immune system with the antibody class switching process and alters the macrophage secretory cytokines profile [388]. Furthermore, captopril modulates the immune response to tumors by increased infiltration of CD3^+^, attenuated tumor-infiltrating CD4^+^, and expression of checkpoint receptor PD-1 on the CD8^+^ cells and CD4 and CD8 double-negative T cells [389]. A novel mechanism through which captopril exerts its antiproliferative and antifibrotic actions is via inhibition of N-acetyl-seryl-aspartyl-lysyl-proline (Ac-SDKP) hydrolysis, a peptide that is known for its immunomodulatory and pro-angiogenic functions [390,391]. As observed, both captopril and Ac-SDKP prevented LV collagen deposition, cellular proliferation, and monocyte/macrophage infiltration.

Overall, captopril demonstrates a range of immunological effects by modulating cytokine expression and immune cell function and interfering with inflammatory signaling cascades. These properties provide support for the integration of captopril into treatment modalities for conditions driven by immune dysregulation beside cardiovascular diseases. Collected findings concerning immunomodulatory effects of captopril are presented in Table 7.

## 4. Conclusions

This review consolidates decades of experimental and clinical evidence in order to prove the remarkable effectiveness of captopril as a multitarget cardioprotective agent. Through its effects on hemodynamic adjustment and neurohormonal suppression, captopril mitigates the adverse morphological and functional alterations associated with MI and HF. Central to its mentioned effects is its ability to reduce both cardiac preload and afterload, preserve LV systolic and diastolic function, and prevent cardiac remodeling processes including myocyte hypertrophy, ventricular dilatation, and changes in fibrous tissue. These effects are not solely mediated through RAAS inhibition but also via antioxidant and anti-inflammatory properties, which are mainly attributed to the presence of the SH moiety.

Numerous aforementioned studies have highlighted its multifaceted mechanisms, which contribute to its efficacy in supporting cardiovascular health, as underscored by its ability to reduce ROS production, scavenge free radicals, and enhance antioxidant defenses. Captopril exhibits modulation of MMP activity and suppression of profibrotic cytokines, thereby influencing critical molecular points included in post-infarction healing and progression toward chronic HF. The variety of these actions emphasizes the need for further studies to deepen our understanding of these antioxidative potentials and clarify the underlying mechanisms through which captopril influences oxidative stress and related pathways, facilitating the identification of novel specific therapeutic interventions.

The evidence gathered from both basic and clinical studies supports captopril’s role in improving survival, exercise tolerance capacity, and long-term cardiac performance. It has also been shown to lead to a delay in disease progression, regardless of whether the treatment was commenced in the early or late stages. Captopril also demonstrated antiarrhythmic effects and preconditioned the myocardium against I/R injury. It exhibited atherosclerosis-reducing properties, suggesting overall protection in the cardiovascular realm. The beneficial effects of captopril have also been evidenced in studies demonstrating its potential to control aging and exhibit antineoplastic properties.

The above-mentioned effects imply promising therapeutic avenues for captopril in addressing a spectrum of health disorders beyond cardiovascular diseases, such as diabetes mellitus, neurodegenerative diseases, and chronic kidney disease. However, conflicting results suggest that the effectiveness of captopril may vary depending on the timing, dosage, and route of drug administration, experimental methodology, and model, as well as the pathophysiological mechanisms involved, which emphasizes the need for a more nuanced understanding of specific therapeutic responses.

## Figures and Tables

**Figure 1 ijms-26-07215-f001:**
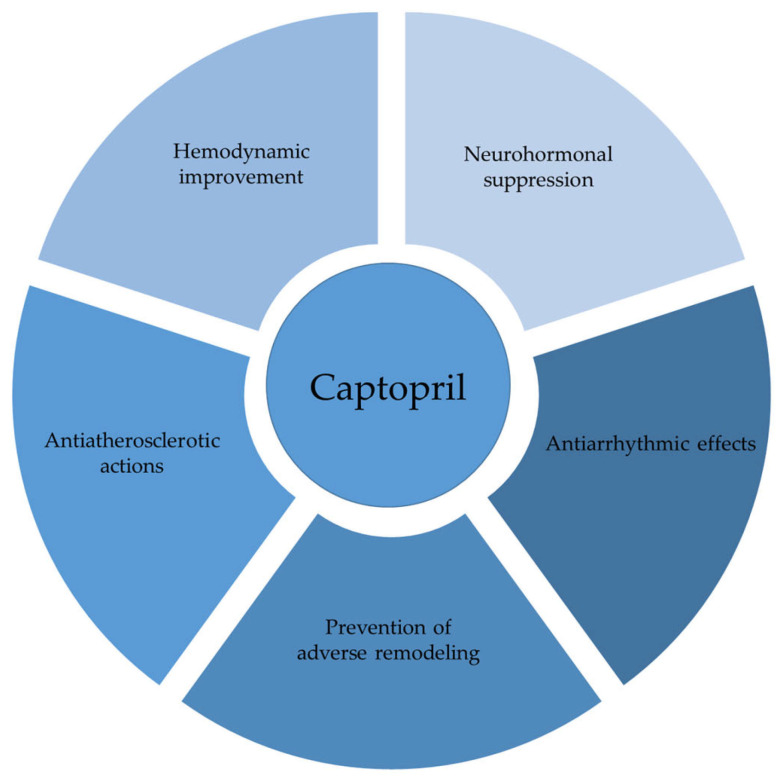
Captopril in cardiovascular modulation.

**Table 2 ijms-26-07215-t002:** Effects of an early and delayed captopril treatment in myocardial infarction models.

References	Timepoint of Administration	Duration of Treatment	Dosage	Results	Comment
[39]	2 h after operation and 21 d after infarction	4 m	2 g/L	Improved hemodynamic function; reduction in the mass of both ventricles and LV dilatation	NS between two timepoints
[56]	Immediately postoperatively	7 d	2 g/L	Unloading systolic and diastolic blood pressure of ventricles; No regression of cardiac hypertrophy	Improved but not restored cardiac pump performance
[57]	On the day of surgery	4–5 w	30 mg/kg i.p.	Reduction in left filling pressures; no change in RV systolic and end-diastolic pressures; reduction in hypertrophic growth in right but not left heart chambers	Minor improvement in systolic LV performance
[58]	After infarction	6 w captopril	30 mg/kg i.p.	Prevention of RVH; reduction in LV weight	Lesser effects of treatment in only early and late therapy
	Early (3 w)	First 3 w captopril, Last 3 w saline		Partial prevention of LVH and no effect on RV weight	
	Late (last 3 w)	First 3 w Saline, Last 3 w captopril		Regression of hypertrophy in both ventricles	
[93]	2nd day of infarction	3 w	500 µg/kg/h s.c.	Reduction in HW; unaltered CO; increased HR and decreased SV and SW	Early therapy had unfavorable effects, yet delayed therapy improved cardiac function
	21st day of infarction	2 w	100 or 500 µg/kg/h s.c.	Decrease in MAP, TPR no reduction in the ventricles’ weight; increased CO through SV	
	5 w from infarction	One-time	10 mg/kg i.v.bolus	Reduced afterload; no change in cardiac performance	
[102]	2nd day of infarction	8 w	2 g/L	Prevention of hypertrophy; improved CI and SV index, reduced SVR	Normalized the myocardial infarction-induced genes expression
[109]	28th day of infarction	28 days	2 g/L	Prevention of progressive myocyte cell lengthening; modest reduction in myocyte cell volume	Even late treatment may attenuate myocyte remodeling
[111]	7th day of infarction	4 w	0.2 g/L	Regression of hypertrophy; preserved or improved EF; reduced ventricular dilatation; improved vascularization in the infarction border zone; no reduction in infarct size and LV anterior wall thickness	Delayed effects are notable

CI—cardiac index; CO—cardiac output; MAP—mean arterial pressure; SV—stroke volume; SVR—systemic vascular resistance; SW—stroke work; TPR—total peripheral resistance; NS—nonsignificant.

**Table 3 ijms-26-07215-t003:** Effects of short-term captopril treatment.

Role-In Group	Daily Dose	Hemodynamic Changes	Effects	Correlation Between PRA and Hemodynamic Changes	Hormonal Changes	Comment	References
NYHA III-IV; n = 10	25–150 mg	CI ↑, MAP ↓, MPAP ↓, MRAP ↓, SVR ↓, MPCWP ↓, HR ↓ *	SV↑, left ventricle filling pressure ↓	No correlation found	Not evaluated	Increased treadmill performance after 4 w	[186]
NYHA III-IV; n = 11	25–275 mg	CI ↑, MAP ↓, RAP ↓, SVR ↓, PVR ↓, PCW ↓, HR↓ *	Arterial and venous vasodilatory effects;	Confirmed for ↓ MAP, SVR, PVR, PCW and ↑CI		Improved treadmill-exercise tolerance after ≥2 m	[187]
NYHA III-IV; n = 10	25–100 mg	CO ↑, SV ↑, SWI ↑, RAP ↓, PAP ↓ SVR ↓, PCW ↓, HR↓	Improved LV function	Not evaluated		Improved exercise duration on the treadmill and bicycle after 8 w	[188]
NYHA III-IV; n = 10	25–125 mg	CI ↑, SVI ↑, SWI ↔, MAP ↓, MPAP ↓, MPCWP ↓, MRAP ↓, SVR ↓, TPuR ↓, HR ↔	Enhanced cardiac performance	Pretreatment PRA correlated with the CI	Aldosterone ↓, Norepinephrine ↑	Increased treadmill-exercise time after 1 m	[189]
NYHA III-IV; n = 5	150–450 mg	CO ↑, MAP ↓, PAP ↓, PCW ↓, RAP ↔	Improved circulatory dynamics	Confirmed for MAP, PAP	Aldosterone ↓, Cortisol ↔, Noradrenaline ↔, Adrenaline ↔	Improved exercise tolerance on subsequent mobilization in the hospital	[193]

CI—cardiac index; CO—cardiac output; HR—heart rate; MAP—mean arterial pressure; MPCWP—mean pulmonary-wedge pressure; SV—stroke volume; PCW—pulmonary-wedge pressure; MPAP—mean pulmonary artery pressure; SVI—stroke volume index; SVR—systemic vascular resistance; MRAP—mean right atrial pressure; PVR—pulmonary vascular resistance; RAP—right atrial pressure; TPuR—total pulmonary resistance; SWI—stroke work index; ↑ increase; ↓ decrease; ↔ no difference; The symbol (*) denotes that the result is not statistically significant.

**Table 4 ijms-26-07215-t004:** Effects of long-term captopril treatment.

Role-In Group	Daily Dose and Follow-Up	Hemodynamic Changes	Effects	Exercise Capacity	Neurohormonal Changes	Comment	References
NYHA III-IV; n = 18	12.5–50 mg TID; 1–3 m	CI ↑, SVI ↑, MAP ↓, MPCWP ↓, SVR ↓, HR ↔	Improved symptoms and NYHA class	Treadmill-exercise duration improved	Not evaluated	Sustained hemodynamic changes occurred during acute phase	[196]
NYHA III-IV; n = 11	75–200 mg; 2–10 m	CI ↑, MAP ↓, TPR ↓, MTT ↓, HR ↔		Not evaluated	PRA ↑, Aldosterone ↓, Catecholamines ↓ *	No correlation between PRA and hemodynamic changes	[211]
NYHA III-IV; n = 9	37.5–300 mg; 3 m	CI ↑, MAP ↓, TPR ↓, PAP ↓, PCW ↓	symptomatic improvement		Not evaluated during the long-term course	Acute-phase hemodynamic improvements were maintained	[195]
NYHA III-IV; n = 36	75–300 mg; up to 36 m	Evaluated during acute phase only	Improved symptoms and NYHA class	Improved at early follow-up (2–5 m), sustained in late follow-up (11–18 m)		Pre- and post-treatment SWI, post-treatment CI and SVI predicted long-term clinical effects	[208]
NYHA III-IV; n = 19	75–300 mg; 6 m	CO ↑, MTT↓; HR ↓	Clinical improvement	Not evaluated	PRA ↑, Aldosterone ↓, Catecholamines ↓ *	Normalization of MTT best correlated with clinical progress	[209]

CI—cardiac index; CO—cardiac output; HR—heart rate; MAP—mean arterial pressure; MPCWP—mean pulmonary-wedge pressure; MTT—pulmonary mean transit time; PAP—pulmonary arterial pressure; PCW—pulmonary-wedge pressure; PRA—plasma renin activity; SVI—stroke volume index; SVR—systemic vascular resistance; SWI—stroke work index; TPR—total peripheral resistance; ↑ increase; ↓ decrease; ↔ no difference; *—not significant.

**Table 5 ijms-26-07215-t005:** Organ-specific enzymatic activity due to different drug modalities.

Tissue Homogenates		Enzymes and Lipid Peroxidation Levels	Observed Drugs, Dosage, Route of Administration and Duration of Treatment		
			Cap 50 mg/kg/d; via Drinking Water; 12 w [357] Hyd 15 mg/kg/d; Ter 45 mg/kg/d		
			WKY		SHR
Myocardium		Cu/Zn-SOD	Cap, Hyd, Ter ↔		Cap, Hyd, Ter ↑
		Mn-SOD	Cap, Hyd, Ter ↔		Cap, Hyd ↔, Ter ↓
		GSH-Px	Cap ↔, Hyd, Ter ↓		Cap, Hyd, Ter ↓
		Catalase	Cap ↑, Hyd, Ter ↔		Cap ↔, Hyd, Ter ↓
		TBARS	Cap, Hyd, Ter ↓		Cap ↔, Hyd, Ter ↓
Liver		Cu/Zn-SOD	Cap, Hyd ↔, Ter ↑		Cap ↓, Hyd, Ter ↑
		Mn-SOD	Cap, Hyd, Ter ↔		Cap ↓, Hyd ↔, Ter↓
		GSH-Px	Cap, Hyd ↓, Ter ↑		Cap, Hyd ↓, Ter ↑
		Catalase	Cap ↓, Hyd ↔, Ter ↑		Cap ↓, Hyd, Ter ↔
		TBARS	Cap ↓, Hyd, Ter ↔		Cap, Hyd ↔, Ter ↑
Skeletal muscle		Cu/Zn-SOD	Cap, Hyd, Ter ↔		Cap, Hyd, Ter ↔
		Mn-SOD	Cap, Hyd, Ter ↔		Cap, Hyd ↔, Ter ↓
		GSH-Px	Cap, Hyd, Ter ↔		Cap, Hyd, Ter ↔
		Catalase	Cap, Hyd, Ter ↓		Cap, Hyd ↔, Ter ↑
		TBARS	Cap, Hyd, Ter ↔		Cap, Hyd, Ter ↔
**Tissue Homogenates**		**Enzymes and Lipid Peroxidation Levels**	**Observed Drugs, Dosage, and Duration of Treatment**		
			**Cap 50 mg/L;** **Enap 20 mg/L via Drinking Water 4–11 w and 11 w**		
			**CF1-Mice**		
			**4–11 w [351]**	**11 w ** **[352]**	**11 w [320]**
Myocardium		Cu/Zn-SOD	Not evaluated	Cap, Enap ↑	Not evaluated
		Mn-SOD		Cap, Enap ↔	
		Se-GSH-Px		Cap ↔	Cap, Enap ↔
		Catalase		Cap, Enap ↔	Not evaluated
		TBARS		Not evaluated	
		GSSG+GSH			Cap, Enap ↔
		GSSG-Rd			Cap, Enap ↔
Liver		Cu/Zn-SOD	Cap, Enap ↑	Not evaluated	Not evaluated
		Mn-SOD	Cap, Enap ↑		
		Se-GSH-Px	Cap, Enap ↑		Cap, Enap ↑
		Catalase	Cap, Enap ↔		Not evaluated
		TBARS	Not evaluated	Cap ↓, Enap ↔	
		GSSG+GSH		Not evaluated	Cap, Enap ↔
		GSSG-Rd			Cap, Enap↑
Kidney	Cortex	Cu/Zn-SOD	Not evaluated	Cap, Enap ↔	Not evaluated
		Mn-SOD		Cap, Enap ↔	
		Se-GSH-Px		Cap ↔	Cap, Enap↑
		Catalase		Cap, Enap ↔	Not evaluated
		TBARS		Not evaluated	
		GSSG+GSH			Cap, Enap ↔
		GSSG-Rd			Cap ↔, Enap ↑
	Medulla	Cu/Zn-SOD		Cap, Enap ↑	Not evaluated
		Mn-SOD		Cap, Enap ↔	
		Se-GSH-Px		Cap ↔, Enap ↑	Cap ↔, Enap ↑
		Catalase		Cap, Enap ↔	Not evaluated
		TBARS		Not evaluated	
		GSSG+GSH			Cap, Enap ↔
		GSSG-Rd			Cap, Enap ↔
Brain		Cu/Zn-SOD		Cap, Enap ↔	Not evaluated
		Mn-SOD		Cap, Enap ↔	
		Se-GSH-Px		Cap ↔	Cap, Enap ↔
		Catalase		Cap, Enap ↔	Not evaluated
		TBARS		Not evaluated	
		GSSG+GSH			Cap, Enap↑
		GSSG-Rd			Cap, Enap↑
Erythrocytes		Cu/Zn-SOD		Cap ↔, Enap ↑	Not evaluated
		Mn-SOD		Cap, Enap ↔	
		Se-GSH-Px		Cap ↔	Cap, Enap ↔
		Catalase		Cap, Enap ↔	Not evaluated
		TBARS		Not evaluated	Cap, Enap ↓
		GSSG+GSH			Cap, Enap↑
		GSSG-Rd			Cap, Enap↑
Lungs		Cu/Zn-SOD			Not evaluated
		Mn-SOD			
		Se-GSH-Px			Cap, Enap ↔
		Catalase			Not evaluated
		TBARS			
		GSSG+GSH			Cap ↔, Enap↑
		GSSG-Rd			Cap, Enap ↔

Cap—captopril; Hyd—hydralazine, Ter—terazosin; Enap—enalapril; ↑ increase activity; ↓ decrease activity; ↔ no difference.

**Table 6 ijms-26-07215-t006:** Dual role of captopril in oxidative balance.

Antioxidative Properties [270,271,272,273,275,276,279,280,281,283,288,295,296,298,300,301,320,321,324,325,326,327,329,339,341,345,349,350,358,361,362,363,364,365]		Pro-Oxidative Properties [288,289,290,366,367]	
Mechanism	Effects	Mechanism	Effects
Free scavenging activity	Direct neutralization of O_2_, OH·; OCL·; HOCL; H_2_O_2_; ^1^O_2_	Fe^3+^-dependent pro-oxidant activity	Induction of lipid peroxidation
Free recyclable scavenger of glutathione	Maintains and regenerates intracellular GSH levels		Generation of OH· and RS·
Restoring Na^+^ K^+^/Mg^2+^ ATPase and Ca^2+^ Mg^2+^/ATPase levels	Stabilizes ion transport and reduces oxidative stress caused by ionic imbalance	Captopril-mediated iron release from ferritin	Generation of TBA reactive deoxyribose degradation products
Inhibition of enhanced lipid peroxidation	Decreases formation of malondialdehyde (MDA) and TBARS	Cu^2+^-dependent pro-oxidant activity	
Induction of PGI_2_ synthesis	PGI_2_ exerts antioxidant effects		
Sustained NO bioavailability	Increases NO synthetase activity and NO production		
Increases enzymatic endogenous antioxidants	Enhances the activity of SOD, GSH-Px, GSH-Rd, CuZn-SOD, and Mn-SOD		
Enhances non-enzymatic antioxidant defense	Protection against oxidative damage		
Preservation of exogenous antioxidants	Maintains α-Tocopherol levels		
Reverses sulfhydryl oxidation of RyRs	Facilitates Ca^2+^ reuptake		
Potentiation of antioxidative effects	Augmented antioxidative capacity of captopril when concomitantly administered with other substances and drugs		

**Table 7 ijms-26-07215-t007:** Key Molecular Mediators of Captopril-Induced Cardioprotection and Immunomodulation.

Effect	Molecular Mediator	References
Reduction	TNF-α	[29,369,370,372,374,380,382]
	IL-1α	[29,380]
	IL-1β	[373,377,378]
	IL-6	[151,373,374,375,377,388]
	TGF-β	[390]
	Immune-complex deposition (C3, IgG)	[12]
	IFNα	[12]
	IL-8	[377]
	IL-10	[380]
	IL-12	[30,151,380,387]
	IL-18	[380]
	MMP-2	[115,117,118,149,263,265]
	MMP-9	[117,118,149,263]
	MPO	[384,385]
Unaltered	Synthesis of C3 component	[29]
	TGF-β1	[388]
Increase	IL-10	[151,372,373,374,379]
	Ac-SDKP	[390]
	IL-2	[379]
	TGF-β	[151,375]
	IL-22	[376]
	PGE_2_	[225]
	6-keto PGF_1α_, TxB_2_	[285]
	PGI_2_	[298,300]
	PGD_2_	[312]

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
