# Peer review of "Cardioprotective Role of Captopril: From Basic to Applied Investigations"

_ijms, 2025, doi:10.3390/ijms26157215_

Round 1

Reviewer 1 Report

Comments and Suggestions for Authors

The manuscript entitled "Cardioprotective Role of Captopril: From Basic to Applied Investigations" was submitted by Jakovljevic and co-workers as a review.

The authors provide a comprehensive and detailed presentation and discussion of published results to the antioxidant, anti-inflammatory and immunomodulatory effects of captopril.

In general, it is of interest to the field to highlight these additional roles of captopril.

Revision points: 

1.) The review would benefit from overview figures:

1.1) One figure, possibly in the introduction section, could provide a summary of the known functions of captopril in the known context of its cardioprotective effects.

1.2) The second figure, possibly in the final part of the review, could summarize the collected finding concerning its newer antioxidant, anti-inflammatory and immunomodulatory functions.

1.3) Possibly connected to the second figure or as extra figure, the authors could highlight the discussed strategies concerning therapies based on the newer findings.

2.) The authors should compare the known aspects of the molecular mechanisms                underlying cardioprotective effects on the one hand and the general immunomodulatory effects on the other hand. 

3.) Table 5: On which references is the table based on ?

4.) Table 6: Is the entire table based on Ref. 363 ?

Author Response

We are grateful to the reviewers for all the useful comments that will significantly improve our manuscript. All corrections in the text are marked with track changes option and colored in red.

Reviewer #1

Revision points: 

1.) The review would benefit from overview figures:

1.1) One figure, possibly in the introduction section, could provide a summary of the known functions of captopril in the known context of its cardioprotective effects.

Answer: Figure 1 representing main cardioprotective effects of captopril is added. Please, see page 3.

1.2) The second figure, possibly in the final part of the review, could summarize the collected finding concerning its newer antioxidant, anti-inflammatory and immunomodulatory functions.

1.3) Possibly connected to the second figure or as extra figure, the authors could highlight the discussed strategies concerning therapies based on the newer findings.

2.) The authors should compare the known aspects of the molecular mechanisms                underlying cardioprotective effects on the one hand and the general immunomodulatory effects on the other hand. 

Answer: In order to fulfill the requests and comments under numbers 1.2, 1.3 and 2, an overview of the immunomodulatory effects of captopril is provided in the final part of the review, along with a tabelar presentation of the key mediator molecules through which captopril exerts its effects. Instead of a figure, we were free to create additional table to make the presentation of the data as well as supporting references clearer in an visual sense. Please, see pages 32-34

3.) Table 5: On which references is the table based on?

Answer: Table 5 summarizes data from almost the entire chapter 3. Role of oxidative stress and circulating molecules in captopril-induced cardioprotection. Numbers of supported references are provided in the table subtitles. Please, page 28.

In order to present all references related to the original Table 5, the table has been moved further down in the text and is now listed as Table 6, to ensure that all relevant citations are properly included. Numbers of supported references are provided in the table subtitles. Please, pages 31-32.

4.) Table 6: Is the entire table based on Ref. 363?

Answer: Supporting references in Table 6 are added. Please, see pages 30 and 31.

The originally designated Table 6 has been moved up in the text and is now presented as Table 5. References related to the data presented in the table have been added. Please, see pages 29-31.

Reviewer 2 Report

Comments and Suggestions for Authors

The article entitled “Cardioprotective Role of Captopril: From Basic to Applied Investigations” deals in a broad and detailed way with the cardioprotective effects of captopril, ranging from basic experimental evidence to clinical studies in humans. The topic is certainly of interest to the scientific community, considering the consolidated role of ACE inhibitors in clinical practice and the potential broader therapeutic implications of captopril, even in non-strictly protected settings. The review is well documented, includes a wide range of studies and presents an in-depth analysis of the physiopathological and molecular mechanisms underlying the effects of the drug. However, the manuscript presents some structural, methodological and stylistic criticalities that currently limit its impact and usability. In particular, the treatment is sometimes excessively verbose and redundant, with passages that could be better summarized, reorganized or focused on the most relevant aspects. In order to support the authors in improving the manuscript, I would suggest inserting a brief statement about the purpose of the review at the beginning because, although the manuscript is rich in details, it lacks a clear introductory section that establishes the main objectives of the review. Also, some sentences in the review seem not very fluid and some paragraphs are too long. I would suggest the authors to review the style for a better flow of the text and use shorter and clearer paragraphs, with adequate spacing.

Author Response

We are grateful to the reviewers for all the useful comments that will significantly improve our manuscript. All corrections in the text are marked with track changes option and colored in red.

Reviewer #2

The article entitled “Cardioprotective Role of Captopril: From Basic to Applied Investigations” deals in a broad and detailed way with the cardioprotective effects of captopril, ranging from basic experimental evidence to clinical studies in humans. The topic is certainly of interest to the scientific community, considering the consolidated role of ACE inhibitors in clinical practice and the potential broader therapeutic implications of captopril, even in non-strictly protected settings. The review is well documented, includes a wide range of studies and presents an in-depth analysis of the physiopathological and molecular mechanisms underlying the effects of the drug. However, the manuscript presents some structural, methodological and stylistic criticalities that currently limit its impact and usability.

  1. In particular, the treatment is sometimes excessively verbose and redundant, with passages that could be better summarized, reorganized or focused on the most relevant aspects.

Answer: Proofreading of the manuscript have been conducted and appropriate language and writting style changes are made throught the manuscript.

  1. In order to support the authors in improving the manuscript, I would suggest inserting a brief statement about the purpose of the review at the beginning because, although the manuscript is rich in details, it lacks a clear introductory section that establishes the main objectives of the review.

Answer: The last paragraph of the section Introduction have been completely modified in order to provide clear purpose/objective of the review. Please, see page 2.

  1. Also, some sentences in the review seem not very fluid and some paragraphs are too long. I would suggest the authors to review the style for a better flow of the text and use shorter and clearer paragraphs, with adequate spacing.

Answer: We made changes mentioned in the first answer and we additionally shortened and separated paragraphs and omitted some part of the text.

Round 2

Reviewer 1 Report

Comments and Suggestions for Authors

The authors have sufficiently addressed my points. The manuscript can be accepted in present form.